# SANSA: Unleashing the Hidden Semantics in SAM2 for Few-Shot Segmentation

**Claudia Cuttano**[*]    **Gabriele Trivigno**[*]    **Giuseppe Averta**    **Carlo Masone**

Politecnico di Torino
{name.surname}@polito.it

## Abstract

Few-shot segmentation aims to segment unseen categories from just a handful of annotated examples. This requires mechanisms to identify semantically related objects across images and accurately produce masks. We note that Segment Anything 2 (SAM2), with its *prompt-and-propagate* mechanism, provides strong segmentation capabilities and a built-in feature matching process. However, we show that its representations are entangled with task-specific cues optimized for object tracking, which impairs its use for tasks requiring higher level semantic understanding. Our key insight is that, despite its class-agnostic pretraining, SAM2 already encodes rich semantic structure in its features. We propose **SANSA** (**S**emantically **A**lig**N**ed **S**egment**A**nything 2), a framework that makes this latent structure explicit, and repurposes SAM2 for few-shot segmentation through minimal task-specific modifications. SANSA achieves state-of-the-art on few-shot segmentation benchmarks designed to assess generalization and outperforms generalist methods in the popular in-context setting. Additionally, it supports flexible promptable interaction via points, boxes, or scribbles, and remains significantly faster and more compact than prior approaches. Code at: https://github.com/ClaudiaCuttano/SANSA.

## 1 Introduction

Segmenting images is a core problem in Computer Vision, yet achieving high-quality results typically requires extensive human effort to annotate pixel-level masks. Moreover, conventional semantic segmentation methods [13, 26, 34, 16] struggle to generalize to unseen categories. Inspired by the human ability to recognize novel objects from just a few examples, few-shot segmentation (FSS) [38, 69, 41, 59] has emerged as a paradigm that leverages a small set of labeled reference samples to guide the segmentation of target images containing arbitrary, previously unseen classes.

To this end, recent work has turned to visual foundation models (VFMs) [7], which offer rich visual representations and strong generalization capabilities [52, 82]. A natural approach [42, 83, 62] is to decouple the few-shot segmentation task into two stages: feature matching followed by promptable segmentation. This is typically achieved by combining DINOv2 [50], known for its strong semantic correspondence capabilities [86, 44, 85, 63], with Segment Anything [35], which excels at producing high-quality segmentation masks [35, 90, 87]. While effective, these modular approaches add computational overhead and require prompt engineering to coordinate multiple VFMs [89].

We observe that Segment Anything 2 (SAM2) [54] offers an alternative paradigm. Designed for Video Object Segmentation, it operates as a *prompt-and-propagate* framework, where an object is specified via its mask and tracked across frames. To achieve this, SAM2 introduces a `Memory Attention` mechanism to implicitly match features across video frames and propagate masks over time with high spatial precision. While this feature matching is originally intended for object tracking

---

[*]Equal contribution.

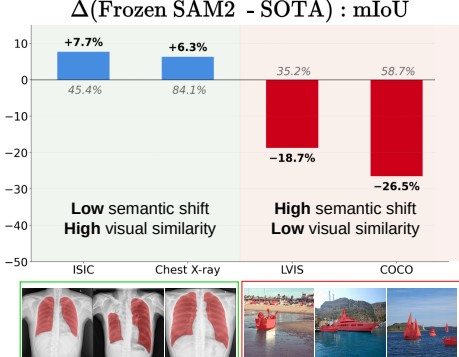

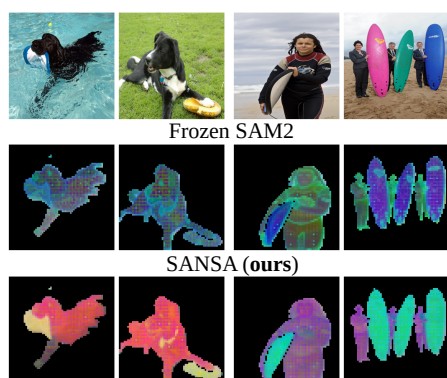

Figure 1: We evaluate frozen SAM2 on few-shot segmentation tasks on four datasets with varying degrees of semantic variability. On datasets [10, 18] with low semantic shift and high intra-class visual similarity, SAM2 matches or even outperforms state-of-the-art APSeg [28]. However, on more challenging datasets like COCO and LVIS, with high semantic shift (*e.g.*, *cruise ship* vs. *rowboat*), its performance drops significantly, compared with GF-SAM [83]. The bottom row illustrates examples of ground-truth masks in both scenarios.

Figure 2: We extract SAM2 features from object instances across diverse images and visualize their distribution using the first three principal components of PCA, mapped to RGB channels. The features appear entangled, with clusters mixing across categories, highlighting the lack of a coherent semantic structure in the original feature space. After adapting the feature space with SANSA, well-defined clusters emerge: semantically similar instances group together, forming coherent structures despite intra-class variation in visual appearance.

based on visual similarity, we note that this architecture inherently unifies two capabilities central to FSS *within a single model*: dense feature matching and high-quality mask generation. Building on this analogy, we propose repurposing the *prompt-and-propagate* framework to address the FSS task, by reinterpreting the temporal dimension of videos as a collection of semantically related images.

This raises the central question: *instead of limiting* SAM2 *to **object tracking**, can it generalize beyond visual similarity to perform **semantic tracking** based on shared concepts across images*?

Addressing this inquiry means, in turn, to answer whether SAM2 has implicitly learned semantic-aware representations, despite its class agnostic pre-training. To answer this, we conduct a toy experiment in Fig. 1, testing SAM2 performance on FSS datasets with different semantic variability. Interestingly, we observe that in low-semantic-shift scenarios, SAM2 achieves comparable or even higher performance with respect to state-of-the-art methods. However, on challenging datasets with high semantic shift, its performance drops drastically. A straightforward conclusion from these preliminary results would be that SAM2 may not have learned discriminative representations in its class-agnostic pretraining, making it unsuited for tasks requiring semantic alignment.

We challenge this interpretation, based on the observation that SAM2 pretraining, focused on instance matching across frames, shares similarities with self-supervised learning frameworks [77, 76, 29, 5], which are known to elicit semantic understanding by enforcing feature invariance across views [11, 4]. Given this analogy, we posit that SAM2 *does* encode semantic information, which is however entangled with instance-specific features optimized for object tracking, reflecting experimental evidence in Fig. 1. If our intuition is true, it implies that *i)* this structure could be disentangled through lightweight transformations [1, 32, 37], such as adapter modules and *ii)* it should be learnable from a set of base classes and generalize to unseen categories [57, 17]. To this end, we *i)* intentionally use one of the simplest adapters in the literature, namely AdaptFormer [31], and *ii)* verify the generalization hypothesis through extensive experiments in strict few-shot benchmarks.

Building on this insight, we introduce **SANSA** (**S**emantically **A**lig**N**ed **S**egment**A**nything 2), and show how to expose SAM2 latent semantic structure, repurposing its `Memory Attention` mechanism to shift from *visual similarity* to *semantic similarity*. The effectiveness of SANSA is illustrated in Fig. 2, where a 3D PCA, computed on *unseen* classes, reveals the emergent semantic organization of our features. We complement our method with a novel training objective designed to exploit SAM2 temporal continuity to convert each target image into a *pseudo-annotation* for subsequent frames.

Our contributions are the following:

- We are the first to investigate the semantic structure within SAM2. We show that such semantics can be disentangled through bottleneck transformations, enabling a unified approach that reinterprets few-shot segmentation as the task of tracking semantic concepts across images;
- We validate SANSA through extensive experiments, achieving SOTA performance in strict FSS benchmarks while also outperforming generalist approaches in the 'in-context' scenario. Our experiments reveal that SAM2 encodes coarse-to-fine semantics, from high level concepts (*e.g.* *dog* vs. *cat*, +6.3% on COCO-20$^i$) to fine-grained distinctions at category-level (*e.g. dalmatian* vs. *bulldog*, +8.3% on LVIS-92$^i$) and part-level (*e.g. hand* vs. *arm* +4.6% on Pascal-Part);
- By supporting prompts like points, boxes, or scribbles, our approach enables a wide range of downstream task, such as data annotation without the need for costly pixel-level masks. Finally, by exploiting the tight integration of memory attention and mask decoding in SAM2, we avoid the need for auxiliary models or complex pipelines, setting a new SOTA with a framework more than 3× faster than competitors, and 4-5× smaller in parameter count.

## 2 Related works

**Few-shot segmentation** aims to segment a target image given an annotated reference. Early methods relied on compressed prototype representations [38, 69, 36, 41, 22], later replaced by attention- and correlation-based approaches [68, 84, 47, 46, 30] to better capture pixel-level relationships. More recently, research focused on leveraging the large-scale pretraining and generalization capabilities of vision foundation models [89, 87, 42, 71, 75]. Matcher [42] and GF-SAM [83] utilize a training-free pipeline: DINOv2 extracts correspondences which are used to prompt SAM for segmentation. Similarly, VRP-SAM [62] leverages a frozen SAM with an external encoder for feature matching. Recent works [87, 89, 44] explore the use of *a single VFM* [7] to jointly handle semantic understanding and mask prediction. DiffewS [89] exploit the emergent semantic correspondences in StableDiffusion [56] to unify the process, while SegIC [44] combines a frozen DINOv2 with a lightweight segmentation decoder. Yet, recent findings [86, 85] suggest that diffusion and DINOv2 features offer complementary but disjoint strengths: the former offers spatial precision but weak semantics, the latter strong semantics but sparse matches. In contrast, we posit that SAM2 unifies both properties: its features possess high spatial granularity and implicitly encode semantic information. We show that this latent semantic structure can be extracted, effectively enabling FSS within a unified architecture.

**Semantic correspondences and Foundation Models.** Finding correspondences between images is a longstanding problem in Computer Vision [6, 43, 33, 55, 49]. While early deep-learning approaches train dedicated models to establish semantic correspondences [33, 55], recent works [86, 63] have shown that VFMs enable generalization across tasks [66, 2, 73, 19]. Among them, DINOv2 [50] and StableDiffusion [56] have demonstrated a compelling ability to establish semantic correspondences between images [86, 49, 85]. Recently, Segment Anything 2 [54] established itself as a foundation model for Video Object Segmentation. We observe that its pretraining, entailing matching object instances across frames under viewpoint changes and motion blur, closely parallels self-supervised learning frameworks [77, 76, 29] that derive semantic understanding through self-similarity training. However, the extent to which SAM2 embeddings encode (if any) semantic concepts has not been studied yet. Recent applications in specialized domains [91, 3] utilize a frozen SAM2 in low-semantic-shift scenarios (*e.g.* propagating masks across slices of 3D imagery given a support example). However, frozen SAM2 shows poor performance in standard FSS benchmarks requiring higher level semantic understanding. In this work, we shed light on this matter, providing insights into SAM2 feature structure and showing that semantic content can be disentanged from its embeddings.

## 3 Method

We structure our investigation around the following key research questions: (1) Can semantic information be effectively extracted from SAM2 features? (2) Can this extraction occur without impairing the functionality of the SAM2 decoder, thereby maintaining its precise segmentation performance? (3) Finally, if a mapping that enhances semantic structure within SAM2 features can be learned, does this mapping generalize effectively to unseen classes?

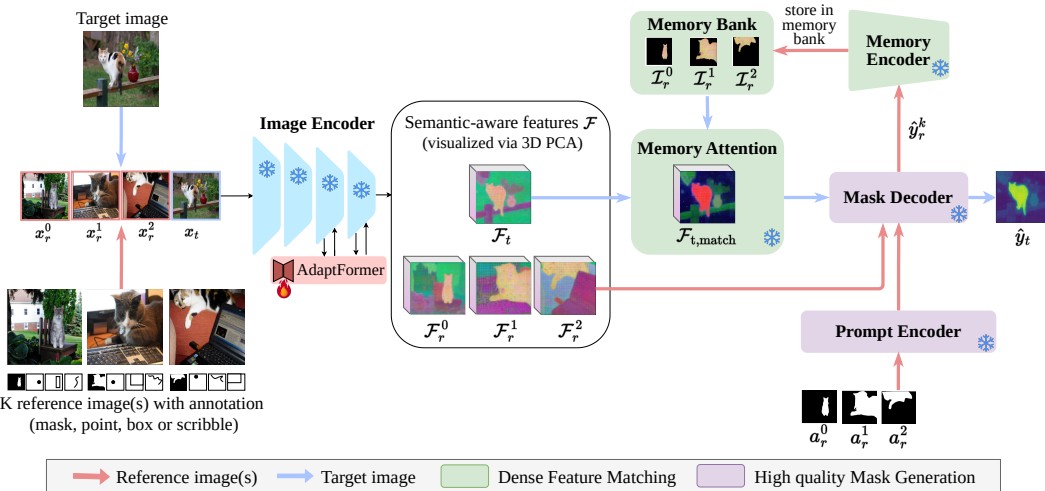

Figure 3: Overview of **SANSA**: Given $k$ annotated reference images and a target image, we construct a pseudo-video by concatenating them, then leverage SAM2 streaming pipeline to process reference frames together with their annotations sequentially. We restructure SAM2 feature space to make its latent semantic structure *explicit*, enabling mask propagation based on *semantic similarity* from reference to target. The emergent semantic structure is visualized by the 3D PCA projection of $\mathcal{F}$.

These questions directly shape the design of our approach. Throughout the rest of the manuscript, we provide empirical evidence supporting affirmative answers to each.

**Task Definition.** We consider the general $k$-shot segmentation setting, where the model is given $K$ reference pairs $R = (x_r^k, a_r^k)_{k=1}^{K}$, each consisting of an image $x_r^k \in \mathbb{R}^{H \times W \times 3}$ and its mask annotation $a_r^k \in [0,1]^{H \times W}$. Given a target image $x_t \in \mathbb{R}^{H \times W \times 3}$, the goal is to predict $y_t \in [0,1]^{H \times W}$, the segmentation mask of objects in $x_t$ that are semantically aligned with the reference. As shown in Fig. 3, we interpret the references and the target as a sequence of frames in a pseudo-video $\mathcal{M}$:

$$\mathcal{M} = [x_r^k, a_r^k]_{k=1}^{K} \cup [x_t, \varnothing], \tag{1}$$

where only the $k$ reference frames are annotated.

## 3.1 From Object Tracking to Semantic Tracking with SAM2

Segment Anything Model 2 extends SAM [35] to Promptable Video Object Segmentation. Like its predecessor, SAM2 comprises three main elements: an `Image Encoder`, a `Prompt Encoder`, and a `Mask Decoder`. Specifically, given an image $x_r^k$ with features $\mathcal{F}_r^k$, and a prompt $a_r^k \in \{\texttt{mask}, \texttt{point}, \texttt{box}\}$, the `Mask Decoder` processes them to produce the segmentation mask $\hat{y}_r^k$. The key innovation of SAM2 lies in its extension to video domain: masks can be propagated across new unannotated frames $x_t$ **without additional prompts**, thanks to a memory mechanism. As shown in Fig. 3, we conceptually decompose SAM2 architecture into two functional components:

- **Dense Feature Matching**: Comprising the `Memory Encoder`, `Memory Bank`, and `Memory Attention`, this module establishes dense correspondences across frames. Concretely, given an object reference mask $\hat{y}_r^k$ (either predicted or given as prompt), the `Memory Encoder` constructs its memory representation, by fusing the mask with frame features $\mathcal{F}_r^k$:

$$\mathcal{I}_r^k = \mathcal{F}_r^k + \texttt{conv\_down}(\hat{y}_r^k). \tag{2}$$

This representation is stored in the `Memory Bank`. Subsequently, the features $\mathcal{F}_t$ of target unannotated frames undergo a cross-attention (`Memory Attention`) that aims at establishing dense correspondences between the current frame and the memory representations from previous frames:

$$\mathcal{F}_{t,\texttt{match}} = \text{Attention}\big(Q(\mathcal{F}_t)K([\mathcal{I}_r^0, ..., \mathcal{I}_r^k])^T\big)V([\mathcal{I}_r^0, ..., \mathcal{I}_r^k]), \tag{3}$$

where $\mathcal{I}_r^k$ are past representations and $Q(\cdot), K(\cdot), V(\cdot)$ the query, key, and value projections.

- **High-quality Mask Generation**: For unannotated frames, the `Mask Decoder` is tasked with refining the coarse features matches $\mathcal{F}_{t,\texttt{match}}$, which encode dense correspondences with prior object representations, to produce the segmentation output $\hat{y}_t$ (*cf* Fig. 3).

We propose to repurpose the memory-based feature matching and mask decoding mechanisms, reinterpreting the temporal dimension of videos as a collection of semantically related images. Thus, rather than tracking a *specific object* across continuous frames, we aim at tracking its *semantic class*.

We highlight two advantages of this formulation: first, unlike recent approaches [62, 44, 89], our model naturally supports variable $k$-shots without modifications; second, our solution seamlessly supports promptable FSS, where prompts can take the form $a_r^k \in \{\texttt{mask, point, box, scribble}\}$, removing the reliance on pixel-level annotations. Finally, we note that reference frames are encoded in `Memory Bank` without undergoing `Memory Attention` (*cf* Fig. 3). By avoiding cross-referencing, we ensure predictions for the target image to be invariant to the ordering of reference images.

## 3.2 SAM2 Feature Adaptation

At the core of the feature matching mechanism lie the learned feature representations, central to the `Memory Attention` mechanism: the reference object representation $\mathcal{I}_r$ is constructed from $\mathcal{F}_r$ (Eq. (2)), and matching is performed via cross-attention between target features $\mathcal{F}_t$ and $\mathcal{I}_r$ (Eq. (3)).

Our goal is repurposing the `Memory Attention` to shift from instance-level to semantic-level matching. To achieve this, we introduce minimal architectural changes and instead focus on restructuring the feature space itself. Specifically, we seek to induce a semantic organization of the features, enabling dense correspondences to reflect **semantic similarity** rather than **visual-similarity**.

We hypothesize that SAM2 features already encode semantic concepts, albeit entangled with signals specific to tracking, such as instance-level details and spatial biases. If such structure exists, then it should be learnable from a set of base classes by training few parameters [1, 32, 57]. Thus, we opt for simple AdaptFormer [14] blocks, although our analysis is not tied to the adaptation method, as we will show in the Experimental Section. We integrate AdaptFormer blocks within the last two layers of the `Image Encoder`, as these encode higher-level semantic representations. Given down- and up-projection matrices $\mathbf{W}_{down} \in \mathcal{R}^{d,\tilde{d}}, \mathbf{W}_{up} \in \mathcal{R}^{\tilde{d},d}$, an AdaptFormer block operates token-wise:

$$\mathcal{A}(x) = \sigma(x \cdot \mathbf{W}_{down}) \cdot \mathbf{W}_{up}, \tag{4}$$

where $\sigma$ is a ReLu and $\tilde{d} < d$ is the bottleneck dimensionality. The adapted features are summed in a residual fashion in the backbone transformer blocks:

$$x_{\text{self}} = \text{Attention}(x), \tag{5}$$
$$x' = \text{MLP}(x_{\text{self}}) + x_{\text{self}} + \mathcal{A}(x_{\text{self}}), \tag{6}$$

The backbone weights are kept frozen and we only train projections $\mathbf{W}_{down}$ and $\mathbf{W}_{up}$.

## 3.3 Training objective

Following standard practice in FSS, we adopt an episodic training paradigm [67, 59, 69, 36]. We have access to a set of training episodes, each one containing annotated instances from a single class. We denote these episodes as $\{x^i, y^i\}$, where $y^i \in [0,1]^{H \times W}$ is the mask that segments these instances.

Leveraging our model inherent ability to process sequences of variable length, we create challenging training examples by inverting the standard $k$-shot setup: instead of predicting a single target image from multiple references, the model receives a single labeled reference image and is tasked with propagating the concept to *multiple unlabeled target images*. Formally, we define the training clip:

$$\mathcal{M}_{train} = [x_r, a_r] \cup [x_t^j, \varnothing]_{j=1}^J, \tag{7}$$

where $\{x_r, a_r\}$ is the annotated reference image and $\{x_t^j\}_{j=1}^J$, are the $J$ unlabeled target images. We feed $\mathcal{M}_{train}$ to our model, which propagates the provided concept across target frames to predict the masklet $\{\hat{y}_t^j\}_{j=1}^J$. Initially, the `Memory Bank` is populated with the reference representation $\mathcal{I}_r$. We propose to condition the prediction for each target frame on the reference as well as on **previous target predictions**, by encoding in the `Memory Bank` the predicted representation $\mathcal{I}_t^j$, computed as

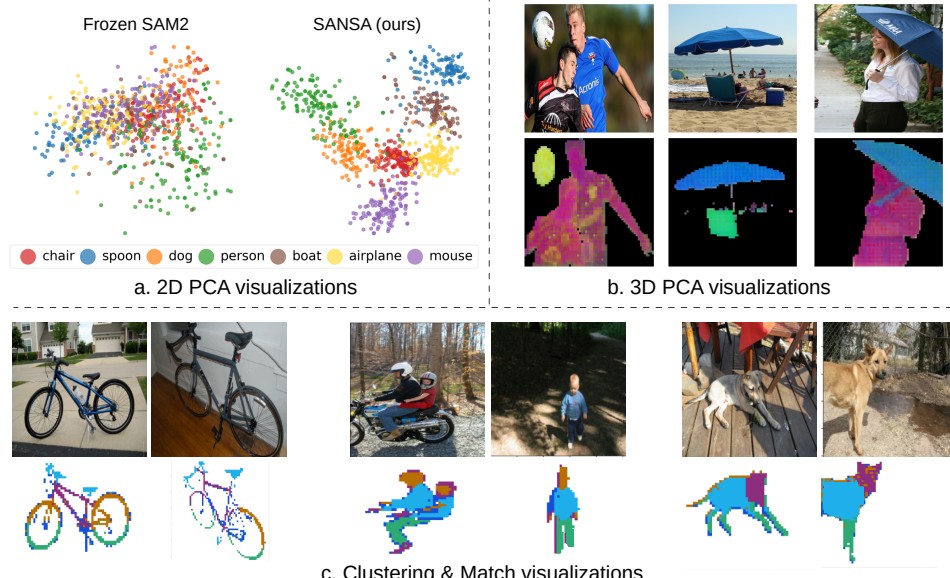

Figure 4: **Semantic structure of feature space.** (a) PCA visualization of frozen SAM2 and our SANSA features on a COCO fold with *unseen* classes, showing the first two principal components color-coded by class. SAM2 features exhibit weak semantic separability, indicating entanglement with other signals. (b) PCA-based RGB visualization of SANSA features across images with *seen* and *unseen* categories, showing consistent semantic mapping. (c) Part-level semantics and cross-image consistency. We cluster features per object and match clusters across image pairs via Hungarian Matching. This reveals that SANSA captures fine-grained distinctions (e.g., *handlebar* vs. *wheel*), spatial layout (e.g., *upper* vs. *lower wheel*), and produces representations that align across images.

in Eq. (2). This design transforms each intermediate frame into a *pseudo-reference* for subsequent frames. This objective discourages overfitting to individual image pairs and encourages robust, semantically grounded correspondences, forcing the model to disentangle semantics from low-level features. We supervise the predicted $\{\hat{y}_t^j\}_{j=1}^J$ with a Binary Cross-Entropy loss and a Dice loss [45].

## 3.4 Visualizing SANSA Feature Space

In this section we analyze how adaptation reshapes SAM2 feature space. In Fig. 4a, we extract features for SAM2 and SANSA for a set of objects belonging to *unseen* classes for our trained model, and visualize their first two Principal Components (PCs), labeled by class. Frozen SAM2 features show weak class discriminability, reinforcing the hypothesis that semantic structure is entangled with task-specific features tailored for the original tracking objective, explaining SAM2 poor FSS performance. Consequently, the leading PCs reveal a mix of semantic and non-semantic signals. In contrast, our SANSA features show clear semantic discriminability, mapping novel classes into well-defined semantic clusters. In Fig. 4b, we visualize SANSA features using PCA, computed jointly across images, where the first three PCs are mapped to RGB channels. The visualization spans three images containing instances from both *unseen* (e.g., person, chair, ball) and *seen* (e.g., umbrella) categories. The consistent color mapping across instances of the same category highlights strong semantic grouping, mapping similar concepts coherently in feature space despite visual variability.

Finally, in Fig. 4c, we shift from coarse category level to fine-grained correspondences at the part level. Following [86], we cluster object features with *K*-Means and match centroids across image pairs via the Hungarian Algorithm, then visualize matched clusters with shared colors to assess semantic consistency. The results show that, despite not being trained with part-level supervision, SANSA features encode part-level understanding (*e.g. hand vs arm*, *handlebar vs wheel*), spatial layout (*e.g. upper vs lower* wheel), and produce representations that align across images.

We provide an in-depth study of semantic representations in Appendix B, using PCA, clustering, and linear probing to analyze how semantics are encoded in SAM2 and made explicit through adaptation.

Table 1: **Strict Few-Shot Segmentation setting**. Results for $k$-shot segmentation on LVIS-92$^i$, COCO-20$^i$, and FSS-1000. We include both specialist models and approaches based on foundation models, trained and tested on disjoint classes. Training-free methods are also reported for reference.

| Method | Venue | Backbone | Params | few-shot segmentation | | | | | |
| | | | | LVIS-92$^i$ | | COCO-20$^i$ | | FSS-1000 | |
| | | | | 1-shot | 5-shot | 1-shot | 5-shot | 1-shot | 5-shot |
|---|---|---|---|---|---|---|---|---|---|
| *Training-free methods* | | | | | | | | | |
| SAM 2 [54] | - | SAM2-L | 224 M | 16.5 | 26.3 | 32.2 | 44.2 | 73.0 | 84.3 |
| PerSAM [87] | ICLR'24 | SAM-H | 641 M | 11.5 | - | 23.0 | - | 71.2 | - |
| Matcher [42] | ICLR'24 | DINOv2-L+SAM-H | 945 M | 33.0 | 40.0 | 52.7 | 60.7 | 87.0 | 89.6 |
| GF-SAM [83] | NeurIPS'24 | DINOv2-L+SAM-H | 945 M | 35.2 | 44.2 | 58.7 | 66.8 | 88.0 | 88.9 |
| *Strict $k$-shot segmentation* | | | | | | | | | |
| AMFormer [72] | NeurIPS'23 | ResNet101 | 49 M | - | - | 51.0 | 57.3 | - | - |
| HMNet [75] | NeurIPS'24 | RN50 | 39 M | - | - | 52.1 | 58.9 | - | - |
| VRP-SAM [62] | CVPR'24 | RN50 + SAM-H | 670 M | 28.3 | - | 53.9 | - | 87.7 | - |
| SegIC [44] | ECCV'24 | DINOv2-G | 1.2 B | 40.5 | - | 53.6 | - | 88.5 | - |
| DiffewS [89] | NeurIPS'24 | StableDiffusion | 890 M | 33.9 | 43.7 | 52.2 | 60.7 | 90.2 | 90.6 |
| **SANSA** | NeurIPS'25 | SAM2-L | 234 M | **48.8** | **53.9** | **60.2** | **64.3** | **91.4** | **92.1** |

# 4 Experiments

**Implementation details.** We employ SAM2 with Hiera-Large [58] as encoder. AdaptFormer [14] is inserted into the last two blocks, with hidden size set to $0.3\times$ the block channel dimension in the strict few-shot setting and $0.8\times$ in the generalist. SAM2 is frozen, and **only the adapters are trained** ($\sim$10M params in the strict case and $\sim$25M in the generalist). We train with AdamW and learning rate $10^{-4}$ for 5 epochs (strict) and 20 (generalist), with $k=1$ (a single annotated reference) and sequence length $J=3$. The same model is evaluated on 1-shot and 5-shot. Full details are in Appendix H.

**Datasets. COCO-20$^i$** [48] is built on MSCOCO [39] and consists of 80 classes split into four folds, each with 20 classes. **FSS-1000** [23] contains 1000 classes, with 520 for training, 240 for validation, and 240 for testing. **LVIS-92$^i$** [42] is more challenging, selecting 920 classes from LVIS [24], divided in 10 folds. **PASCAL-Part** [42] includes four superclasses with 56 object parts across 15 classes. **PACO-Part** is built from PACO [53] and contains 303 classes, split in four folds.

## 4.1 Strict Few-Shot Segmentation Setting

We first evaluate our method in strict FSS setting. Following standard protocols [72, 62, 61], we train on base classes and evaluate on **novel** classes with $k$-shots. These experiment address the question of whether the *semantic mapping* learned on base classes can transfer meaningfully to novel categories.

**Few-shot segmentation.** In Tab. 1, we compare SANSA, besides specialist models, such as AM-Former [72] and HMNet [75], with the most relevant and recent approaches based on Foundation Models. These include methods that, like ours, leverage a single Foundation Model, such as SegIC [44] and DiffewS [89], as well as modular two-stage pipelines like VRP-SAM [62]. For completeness, we also report results from generalist *training-free* methods built on DINOv2 and SAM, including PerSAM [87], Matcher [42], GF-SAM [83], as well as our baseline, *i.e.* frozen SAM2.

In the one-shot setting, SANSA consistently outperforms all prior methods, demonstrating superior generalization to unseen classes. Specifically, we surpass the best direct competitors SegIC, VRP-SAM and DiffwS by +8.3%, +6.3%, and +1.2% on LVIS-92$^i$, COCO-20$^i$, and FSS-1000, respectively. This performance gap remains consistent also against training-free approaches: compared to GF-SAM, SANSA achieves gains of +13.6%, +1.5%, and +3.4%. On the challenging LVIS-92$^i$, including 920 fine-grained categories (e.g., *bulldog*, *dalmatian*), DINO-based methods (SegIC, GF-SAM, Matcher) tend to underperform, possibly due to DINO overly-semantic features [20, 86] (*e.g.*, grouping distinct breeds under the concept of "dog"). Here, SANSA achieves a substantial +8.3% gain, suggesting that SAM2 encodes a latent hierarchical semantic structure, capturing both high-level semantic concepts (dominant in COCO-20$^i$) and fine-grained distinctions (crucial for LVIS-92$^i$).

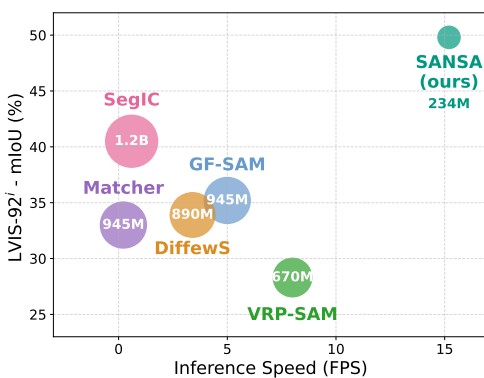

| | VRP-SAM | SANSA(ours) | |
|---|---|---|---|
| Params | 670M | 234M | |
| *point* | 38.4 | 53.4 | +15.0 |
| *scribble* | 47.3 | 53.1 | +5.8 |
| *box* | 49.7 | 54.3 | +4.6 |
| *mask* | 53.9 | 60.2 | +6.3 |

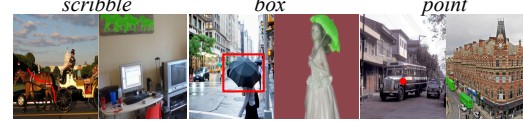

Figure 5: **Comparison of inference speed and mIoU**, with bubble size representing #parameters. The plot highlights our superior trade-off.

Figure 6: **Promptable few-shot segmentation with SANSA.** Top: performance in strict few-shot (COCO-20$^i$) using different prompt types compared with VRP-SAM. Bottom: qualitative examples with point, scribble, and box prompts.

Regarding the 5-shot setting, we note that SegIC and VRP-SAM do not provide an inference pipeline to extend to $k$-shot. In contrast, SANSA natively models correspondences across multiple reference frames and it outperforms the best competitor, DiffewS, by +10.2% and +3.6% on LVIS-92$^i$ and COCO-20$^i$, respectively. Compared to training-free models, SANSA shows improvements of +9.7%, +3.2%, on LVIS-92$^i$ and FSS-1000, while suffering a -2.5% gap on COCO-20$^i$ w.r.t. GF-SAM. We also report results of upgrading SAM-based baselines to SAM2 in Appendix D.

**Performance-Efficiency Trade-off.** In Fig. 5, we analyze the trade-off between model size, inference speed (img/s), and performance. SANSA achieves state-of-the-art results while being the most lightweight solution. Specifically, it is: *i)* over 3× faster than the direct competing method (GF-SAM), *ii)* more compact, introducing only adapter parameters on top of SAM2 (totaling 234M), and *iii)* substantially more accurate, outperforming GF-SAM by +13.6% mIoU on the challenging LVIS-92$^i$ benchmark. Importantly, SANSA keeps the SAM2 architecture entirely frozen, meaning that by storing only the adapter weights, it retains SAM2 state-of-the-art performance on Video Object Segmentation while also achieving top-tier results on few-shot segmentation within a single model. A large-scale annotation study quantifying speed/quality trade-offs is presented in Appendix C.

**One-Shot Promptable Segmentation.** Our framework keeps SAM2 decoder frozen, maintaining its capability to generate masks from any prompt (*i.e.* points, scribbles, or boxes). As such, during inference, users can segment reference objects by providing a simple point, without requiring costly pixel-level masks. In Fig. 6 we evaluate the performance with different prompts on strict FSS setting on COCO-20$^i$ against VRP-SAM, which also supports such prompts. SANSA shows gains of +15.0%, +4.6%, and +5.8% when using points, scribbles, and boxes as annotations. More importantly, the performance drop from masks to point prompts is substantially smaller for SANSA (-6.8%) compared to VRP-SAM (-15.5%). We attribute this to the tightly coupled design of our architecture, which maximizes feature reuse by jointly leveraging the same representations for prompt encoding, feature matching, and mask decoding. Prompt generation details and qualitative results are in Appendix E.

### 4.2 Generalist In-Context Setting

Following recent few-shot segmentation works [44, 89, 42, 83], we evaluate SANSA in the in-context segmentation setting, where a single generalist model is tested across multiple benchmarks (*cf.* Tab. 2). Given the lack of a standardized training protocol, we explicitly report additional datasets used by each method beyond ADE20K and COCO, which are shared across all approaches. We evaluate three configurations of our approach: *i)* a minimal setup using only COCO and ADE20K, *ii)* an extended version incorporating LVIS, following the setup of SegIC [44], and *iii)* a variant including PACO to mitigate object-level bias and reinforce part segmentation capabilities.

When trained only on COCO and ADE20K coarse categories, SANSA exhibits strong out-of-domain generalization on the LVIS benchmark, outperforming DiffewS and SINE by +4.8% and +5.0%, respectively. Moreover, when LVIS is included in the training set (matching SegIC in-domain setup), SANSA further improves performance and surpasses SegIC by +5.4%. Despite not being trained at

Table 2: **Performance of SANSA against Generalist In-context models**. Excluding training-free approaches, all methods are trained on COCO and ADE20k, and we report additional training datasets for each one. SegIC (*) uses additional supervision via a textual meta-prompt at test time.

| Method | Additional Datasets | Params | few-shot segmentation | | | | | | part seg. | |
| | | | LVIS-92$^i$ | | COCO-20$^i$ | | FSS-1000 | | Pascal Part | PACO Part |
| | | | 1-shot | 5-shot | 1-shot | 5-shot | 1-shot | 5-shot | | |
|---|---|---|---|---|---|---|---|---|---|---|
| *Training-free methods* | | | | | | | | | | |
| Matcher [42] | *n.a.* | 945M | 33.0 | 40.0 | 52.7 | 60.7 | 87.0 | 89.6 | 42.9 | 34.7 |
| GF-SAM [83] | *n.a.* | 945M | 35.2 | 44.2 | 58.7 | 66.8 | 88.0 | 88.9 | 44.5 | 36.3 |
| Painter [70] | NYUv2 | 354 M | 10.5 | 10.9 | 33.1 | 32.6 | 61.7 | 62.3 | 30.4 | 14.1 |
| SegGPT [71] | Pascal, PACO | 354 M | 18.6 | 25.4 | 56.1 | 67.9 | 85.6 | 89.3 | 35.8 | 13.5 |
| SINE [40] | Object365, PACO | 373 M | 31.2 | 35.5 | 64.5 | 66.1 | - | - | 36.2 | - |
| DiffewS [89] | Pascal | 890 M | 31.4 | 35.4 | 71.3 | 72.2 | 87.8 | 88.0 | 34.0 | 22.8 |
| SegIC [44] | LVIS | 1.2 B | 47.8 | - | 74.5* | - | 88.4 | - | 31.2 | 18.7 |
| **SANSA** | - | 250 M | 36.2 | 42.5 | **76.4** | 77.1 | 88.2 | 89.0 | 40.4 | 29.2 |
| **SANSA** | LVIS | 250 M | **53.2** | 58.1 | 74.7 | 76.3 | 89.1 | 89.7 | 41.5 | 31.2 |
| **SANSA** | LVIS, PACO | 250 M | 50.3 | **59.0** | 75.6 | **78.6** | **90.0** | **91.0** | **49.1** | **43.0** |

the part level, SANSA exhibits strong cross-task generalization capabilities, surpassing generalist baselines by a large margin (+7.5% and +8.4% over DiffewS, the best competitor on Pascal-Part and Paco-Part, respectively), including those trained on part segmentation datasets [71, 40]. Our method is only outperformed by training-free models, which, by design, are not biased by object-level training. To address this bias, we follow [71, 40] and augment our training set with PACO, leading to +6.7% on Paco-Part (in-domain) and +4.6% on Pascal-Part (out-of-domain) against the strong baseline of GF-SAM. Finally, we highlight that, within the generalist model category, SANSA is the most compact solution, with only 250 M parameters.

## 4.3 Generalization across domains and styles

To explore how SANSA generalizes beyond the scope of standard benchmarks, we follow [62, 42] and present qualitative examples drawn from in-the-wild image pairs, shown in Fig. 7. In each case, the model is tasked with few-shot segmentation using a reference image from COCO [39] and a target image collected from the web, offering a complementary view to benchmark evaluations.

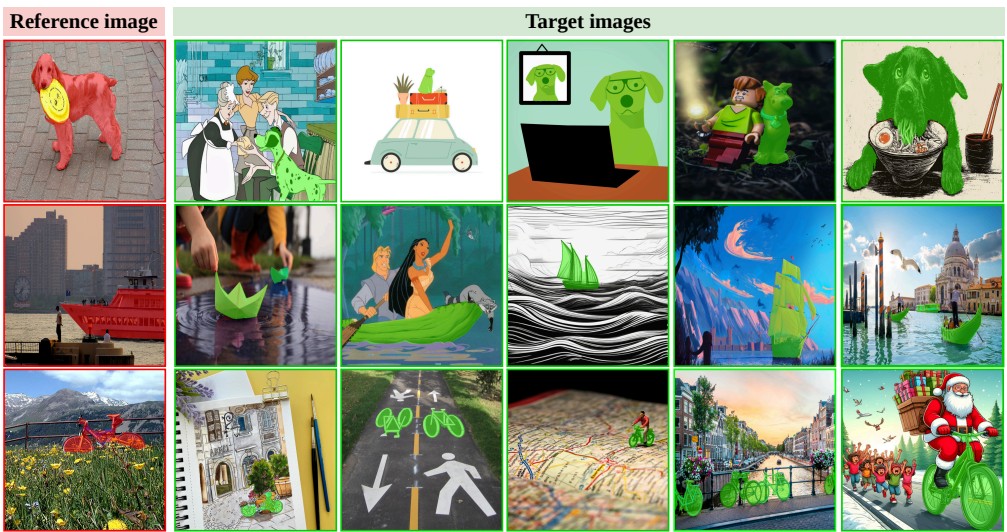

Figure 7: **Few-shot segmentation with SANSA in-the-wild**. These examples showcase SANSA ability to handle alternative types of challenges not typically covered by standard benchmarks, such as domain shifts (*e.g.*, real-world to cartoon) and style variations (*e.g.*, photos to sketches).

These examples introduce a different challenge compared to traditional benchmarks, focusing on domain and style shifts, such as when the target image is a cartoon or stylized sketch, or when the

Table 3: Ablation on COCO-20$^i$. **Left:** fine-tuning underperforms adaptation. **Middle:** simple adapters yield strong gains, while higher capacity (larger bottlenecks or added complexity, *e.g.*, MONA) hurts generalization. **Right:** adapting the last two stages suffices to disentangle semantics.

| Fine-tuning strategies | | | Adaptation strategies | | Adapter placement | | |
|---|---|---|---|---|---|---|---|
| Method | Params | mIoU | Method | mIoU | Stages | | mIoU |
| Frozen | 0 | 32.2 | LoRA [32] | 58.0 | None | – | 32.2 |
| From Scratch | 224 M | 37.1 | Adapter [31] | 59.4 | All | 0–3 | 59.4 |
| Full FT | 224 M | 51.6 | MONA [80] | 56.9 | Early | 0–1 | 38.7 |
| Decoder FT | 4 M | 52.1 | AdaptFormer [14] | | Middle | 1–2 | 57.9 |
| QKV FT | 50 M | 55.3 | Bottleneck $0.8\times$ channel dim | 58.2 | Late | 2–3 | **60.2** |
| Backbone FT | 210 M | 55.2 | Bottleneck $0.5\times$ channel dim | 59.6 | | | |
| SANSA | 10 M | **60.2** | Bottleneck $0.3\times$ channel dim | **60.2** | | | |

visual appearance changes dramatically between the reference and target. These results demonstrate SANSA strong generalization. Importantly, this robustness across domains and styles is inherited from the frozen SAM2. Our method preserves this capability by learning a semantic mapping within the frozen feature space, enabling reliable correspondence even under significant distribution shifts.

## 4.4 Ablation studies

We investigate three central aspects of our design: adaptation vs. fine-tuning, adapter architecture and capacity, and adapter placement. Results are summarized in Tab. 3.

**Why adaptation instead of fine-tuning?** A central question of our work is how to distill knowledge from a pretrained model (*i.e.*, SAM2) while preserving generalization, a key challenge in FSS. To this end, we evaluate fine-tuning strategies targeting the decoder, QKV projections, backbone, and full model, and compare them with inserting adapters into frozen weights. Results show that adaptation outperforms fine-tuning, indicating that it better preserves SAM2 pretrained priors, shifting the embedding space toward task-relevant semantics without altering the underlying representations.

**What makes an adaptation strategy effective?** Basic adapters such as LoRA [32], Adapter [31], and AdaptFormer [14] all yield similar gains around ∼27% mIoU over frozen SAM2. The slight gains (∼2%) with Adapter and AdaptFormer w.r.t. LoRA suggest that a simple non-linearity can refine this structure but not fundamentally reshape it. By contrast, increasing adapter capacity, either by enlarging the bottleneck or using more complex designs such as MONA [80], reduces generalization. These results show that effective adaptation thus requires simplicity and constraint: low-capacity, bottlenecked modules best expose SAM2 latent semantics, while excessive capacity tends to overfit.

**Where should adapters be placed?** Prior works often insert adapters throughout the entire backbone. In our case, we find that adapting only the last two stages is sufficient, as these layers already capture high-level semantic information, which is the focus of our disentanglement objective.

Additional ablations, including backbone scale and training objective, are provided in Appendix D.

## 5 Conclusions

In this work, we introduced SANSA, which enhances SAM2 to accept and propagate a *visual* prompt across frames, and showed that its memory attention mechanism can be re-focused towards semantic correspondences by restructuring the feature space. We addressed three fundamental inquiries: (1) we demonstrated that semantic information can be extracted from SAM2 features through lightweight bottleneck transformations, answering our initial research question, (2) we showed that this can be achieved while keeping SAM2 frozen, thereby preserving its segmentation capabilities, and finally (3) we experimentally verified that the learned semantic mapping generalizes robustly to novel classes, with SOTA performance on strict few-shot benchmarks and against generalist models. Beyond technical contributions, SANSA offers practical advantages: it supports diverse prompts for annotation-efficient applications, and offers a 3x speed increase. Finally, our results suggest that foundation models like SAM2 may contain richer task-adaptable knowledge than their original objectives imply, a direction worthy of future exploration.

**Acknowledgements.** Claudia Cuttano was supported by the Sustainable Mobility Center (CNMS) which received funding from the European Union Next Generation EU (Piano Nazionale di Ripresa e Resilienza (PNRR), Missione 4 Componente 2 Investimento 1.4 "Potenziamento strutture di ricerca e creazione di "campioni nazionali di R&S" su alcune Key Enabling Technologies") with grant agreement no. CN_00000023.

Gabriele Trivigno, Giuseppe Averta and Carlo Masone were supported by FAIR - Future Artificial Intelligence Research which received funding from the European Union Next-GenerationEU (PIANO NAZIONALE DI RIPRESA E RESILIENZA (PNRR) – MISSIONE 4 COMPONENTE 2, INVES-TIMENTO 1.3 – D.D. 1555 11/10/2022, PE00000013). This manuscript reflects only the authors' views and opinions, neither the European Union nor the European Commission can be considered responsible for them.

We acknowledge the CINECA award under the ISCRA initiative, for the availability of high performance computing resources.

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

## Appendix

**Table of Contents:**

## A  Discussion

### A.1  Limitations and Future Works

By keeping the Segment Anything 2 (SAM2) [54] model entirely frozen and storing only adapter weights, our SANSA preserves SAM2 state-of-the-art performance on Video Object Segmentation (VOS), while enabling few-shot segmentation within a unified pipeline. As our method builds directly upon SAM2, it also inherits its design assumptions and limitations. In particular, while SAM2 supports tracking and segmenting multiple objects across a video, it does so by processing each object independently. This behavior is standard in few-shot segmentation, where each object is typically queried and segmented in isolation using a separate prompt. Nevertheless, future work could explore leveraging contextual relationships between multiple objects, which may improve segmentation accuracy through cross-instance interactions and enable multi-class few-shot segmentation.

### A.2  Societal Impacts

Our method leverages the publicly available, pre-trained weights of Segment Anything 2 (SAM2), thus requiring minimal additional computational resources. This significantly limits the environmental impact commonly associated with training large foundation models from scratch. By providing an analysis of the underlying mechanisms of foundation models like SAM2, we contribute to a deeper understanding of their operation and potential applications beyond conventional tasks. We believe this knowledge dissemination can foster broader awareness and acceptance of foundation models both within and outside the scientific community.

Moreover, our approach facilitates the efficient annotation of large-scale datasets with minimal human effort (see Section C). Given that data annotation represents one of the primary bottlenecks and cost drivers in deploying vision models in real-world scenarios, our method could bring benefit in reducing the impact of computational burdens associated with dataset curation.

However, we recognize potential risks inherent to few-shot learning methods, including SANSA. Specifically, its capability for rapid adaptation with limited data could be exploited in ways that raise ethical and privacy concerns. For example, SANSA might be applied in surveillance systems to identify individuals without their consent, posing risks to personal privacy and civil liberties.

## B  Emergence of Semantic Representations

### B.1  Analyzing Semantic Structure via Principal Component Decomposition

Our hypothesis is that SAM2 features *do* encode semantic information, albeit entangled with low-level signals, whereas our adapted SANSA features disentangle this semantic content to make it *explicit*. To validate this hypothesis, we analyze the feature spaces of both models via a clustering-based experiment, reported in Fig. 8. Specifically, we evaluate how well semantic clusters form in lower-dimensional subspaces obtained via PCA, comparing frozen SAM2 to our adapted SANSA features. To this end, we extract object-level features from fold 0 of COCO-20$^i$, which contains

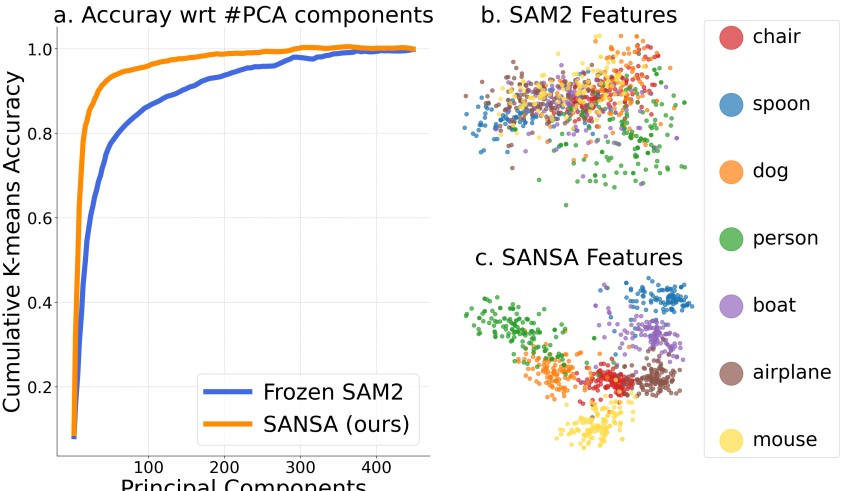

Figure 8: **Semantic information concentration across principal components.** (a) We evaluate how semantic information is distributed across the principal components of the feature space. For each number of retained components (x-axis), we compute class centroids on COCO-$20^i$ training embeddings and assign test embeddings to the nearest centroid. The y-axis shows the relative classification accuracy, normalized by the accuracy at full dimensionality. A steep rise (SANSA) indicates that semantic information is concentrated in the leading components; a gradual rise (SAM2) suggests it is entangled with other signals. (b) 2D PCA projection of frozen SAM2 features, colored by class. Semantic structure is weak, with significant overlap across classes. (c) 2D PCA projection of SANSA features, showing compact and well-separated clusters aligned with semantic categories.

categories unseen by SANSA (trained on the remaining folds). We then analyze the semantic organization of these features through a clustering-based assignment task combined with Principal Component Analysis (PCA):

1. We apply PCA to the extracted features, varying the number of principal components (PCs) retained, from 2 up to 450, capturing 99% of the total variance.

2. For each reduced-dimensionality representation, class centroids are computed on the training set features using $k$-means clustering.

3. During testing, embeddings from the test set are assigned to the nearest centroid based on minimum Euclidean distance.

4. The assignment accuracy per class is computed using the ground-truth labels.

**Plot (a)** reports the relative assignment accuracy (y-axis) as a function of the number of PCs (x-axis). This accuracy is normalized by the overall accuracy obtained with the full feature space (*i.e.*, no dimensionality reduction) for each method, allowing a direct comparison of how semantic information concentrates across principal components independently of absolute accuracy differences. The results reveal a marked difference between SAM2 and SANSA. The blue curve (SAM2) shows a gradual increase in accuracy as more PCs are included, indicating that semantic and non-semantic signals are mixed across many leading components. Conversely, the orange curve (SANSA) exhibits a sharp rise with only a few PCs, revealing that semantic information is concentrated and explicitly encoded in the leading principal components of the adapted feature space.

**Plots (b)** and **(c)** provide a visual illustration of this phenomenon by projecting the features onto the first two principal components and coloring them by class label. The frozen SAM2 features, in plot (b), show weak class discriminability, confirming that their semantic structure is entangled with task-specific features optimized for SAM2 original tracking objective. Conversely, SANSA features, in plot (c), form compact, well-separated clusters, indicating that semantic information dominates the main axes of variation in the adapted feature space.

### B.2 Semantic Discriminability and Downstream Transferability

The previous analysis revealed that in the adapted SANSA feature space, semantic information is predominantly encoded. This raises a natural follow-up question: *to what extent this semantic information was already encoded in the original, frozen* SAM2 *features, and how easily can it be accessed for downstream tasks?*

To investigate this, we present a two-fold analysis in Fig. 9, comparing two models: *i)* frozen SAM2, and *ii)* our SANSA trained on folds $1, 2, 3$ of COCO-$20^i$. We conduct two experiments on fold 0, *i.e.* on *unseen* classes for our model:

1. **Linear probing**: following standard practice in representation learning [25, 50, 11], we conduct a linear probing experiment to assess semantic discriminability in the two feature spaces. We extract features from both models and train a pixel-level **linear** classifier on the training set of COCO-$20^i$ fold 0. Note that both models are frozen and only a single linear layer is trained for the task of semantic segmentation. Evaluation is performed on the corresponding validation set. A high mean Intersection-over-Union (mIoU) indicates that semantic signals are present in the feature space and can be recovered by a simple linear projection.
2. **Few-Shot Segmentation**: on the same COCO-$20^i$ fold, we test the performance of frozen SAM2 and SANSA on unseen classes in the downstream task of few-shot segmentation.

On the left side of the plot, the results indicate that SAM2 achieves 65.8% mIoU with a simple linear probe, confirming that they encode a non-trivial amount of semantic information that is linearly accessible. Interestingly, when repeating the same experiment using our SANSA features, we observe only a modest improvement to 69.2% mIoU (+3.4%).

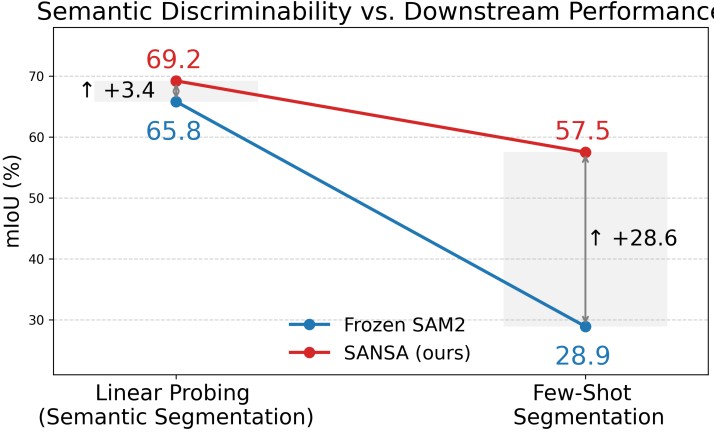

Figure 9: Left: **Linear probing** for Semantic Segmentation on a fold of COCO-$20^i$, comparing frozen SAM2 and SANSA (trained on a disjoint class set). Right: **Few-shot segmentation performance on the same set of data.** Despite poor performance on few-shot segmentation, a simple linear layer (*cf* linear probing) can learn to discriminate semantic-classes from SAM2 frozen features. These results, together, suggest the presence of semantic structure, albeit entangled with other signals.

On the right, FSS results show that SAM2 achieves 28.9% mIoU, while SANSA reaches 57.5%, with a substantial +28.6% improvement. When considered together, these results highlight a key insight: although SAM2 features already encode meaningful semantic cues, their utility in downstream tasks is limited by the way this information is entangled with low-level and instance-specific signals. The relatively modest gain in linear probing performance (+3.4%) compared to the substantial improvement in few-shot segmentation (+**28.6**%) suggests that SANSA primarily reorganizes the latent structure into an *explicitly* semantic form, enabling downstream tasks that require semantic-level understanding.

### B.3 Extracting Semantics Without Compromising SAM2 Capabilities

In the previous sections, we investigate how SANSA exposes the latent semantic structure embedded in frozen SAM2 features, which was our main goal. Moreover, an important principle of our design was to extract this semantic structure without sacrificing SAM2 Promptable Segmentation capabilities or its state-of-the-art performance on Video Object Segmentation. Consequently, our choice was to insert AdaptFormer blocks [14] into the last two layers of the frozen `Image Encoder`, which introduce residual bottleneck projections, reorganizing the feature space without altering any of the SAM2 weights.

For comparison, we evaluate several alternative adaptation strategies: *i)* fine-tuning only the decoder, *ii)* fine-tuning the Query, Key, and Value (QKV) projections within the attention layers [65], and *iii)* full backbone fine-tuning. Decoder fine-tuning updates around 4 million parameters, QKV fine-tuning modifies approximately 50 million parameters, and full backbone fine-tuning involves over 210 million parameters.

Tab. 4 reports results under two settings: out-of-domain, where models are trained on a subset of COCO-$20^i$ folds and evaluated on unseen classes (*i.e.*, the strict few-shot segmentation setting, which is our primary focus), and in-domain, where models are trained on all categories.

Table 4: **Comparison of adaptation strategies**. We compare various strategies for adapting SAM2 in terms of generalization to unseen classes (out-of-domain) and performance on seen classes (in-domain). SANSA achieves the best out-of-domain mIoU with minimal parameter overhead, while preserving the original SAM2 weights.

| Adaptation Strategy | COCO-$20^i$ | | Trainable Parameters | SAM2 Weights Update |
|---|---|---|---|---|
| | in-domain | out-of-domain | | |
| Decoder Finetuning | 63.7 | 52.1 | 4 M | ✓ |
| QKV-only Finetuning | 73.5 | 55.3 | 50 M | ✓ |
| Backbone Finetuning | 74.8 | 55.2 | 210 M | ✓ |
| AdaptFormer (**SANSA**) | 72.0 | 60.2 | 10 M | ✗ |

The table highlights the advantages of our approach. Tuning the decoder provides sub-optimal results both in- and out-of-domain, as performances are constrained by the limited semantic structure of frozen features. Finetuning the backbone, or the attention layers, is effective in producing semantic aware features, as shown by the in-domain performance. However, such fine-tuning, despite its greater capacity which boosts in-domain performance, leads to overfitting to training categories, as shown by the large performance drop on unseen classes.

On the other hand, adapters restrict updates to a low-dimensional bottleneck [32, 31], forcing the model to prioritize generalizable transformations over spurious in-domain correlations [27], demonstrating more consistent and stable performance across both seen and unseen classes.

Finally, by maintaining the SAM2 backbone and decoder entirely frozen, we ensure that the original model core capabilities, such as Promptable Segmentation and Video Object Segmentation, remain uncompromised. This is critical to deploying SANSA in practical settings, obtaining a model that fully supports both geometric and visual prompts, by storing only the adapter weights.

### B.4 Exploring Implicit Semantics in Traditional Video Object Segmentation Methods

Our hypothesis is that the class-agnostic training objective of Video Object Segmentation encourages the learning of semantically meaningful representations, even without category supervision. While our main analysis is focused on SAM2, which benefits from its large-scale pretraining, we investigate whether similar semantic understanding properties arise in traditional VOS trackers trained at smaller scale.

To this end, we evaluate three representative VOS methods, AOT [78], DeAOT [79], and MAVOS [60], alongside SAM2, on the COCO-$20^i$ 1-shot segmentation benchmark. As shown in Tab. 5, all models exhibit relatively weak performance when directly transferred to few-shot segmentation. We then apply our adaptation to each of these models, and evaluate them in the *strict* FSS setting on COCO-$20^i$, *i.e.* focusing on their ability to generalize to previously unseen classes. The results show a

consistent behavior: despite their different training scales and capacities, all VOS models demonstrate a large performance gap between their frozen and adapted versions. For instance, MAVOS improves from 27.1% to 52.3% mIoU after adaptation, while SAM2 with Hiera-B improves from 28.0 % to 55.4%. The gap between these methods and SAM2-B is attributable to its broader generalization capabilities, stemming from its diverse and large-scale pretraining.

Table 5: **Evaluation of traditional Video Object Segmentation methods and SAM2 with and without SANSA adaptation.** We report mIoU (%) on COCO-20$^i$ 1-shot segmentation.

| Method | Backbone | #Params | Frozen | +SANSA |
|---|---|---|---|---|
| AOT [78] | Swin-B | 70 M | 25.0 | 48.5 |
| DeAOT [79] | Swin-B | 75 M | 26.9 | 51.0 |
| MAVOS [60] | Swin-B | 96 M | 27.1 | 52.3 |
| SAM2 [54] | Hiera-B | 86 M | 28.0 | 55.4 |
| SAM2 [54] | Hiera-L | 234 M | 32.2 | 60.2 |

This consistent behavior across models suggests that the VOS training paradigm inherently encourages the emergence of transferable semantic representations. In summary, our findings support the view that such semantic structure is an implicit property of the VOS objective itself, and that it can be effectively made *explicit*, and successfully repurposed, for downstream semantic tasks.

## C   Scalable and Fast Annotation

While the previous sections have established the strong segmentation performance of SANSA across benchmarks, a particularly compelling advantage of our approach is its ability to facilitate fast and scalable annotation in practical, real-world scenarios. In many applications, such as constructing new datasets or adapting models to novel domains, rapid annotation of large image collections without the need for fine-tuning on test categories is essential.

To rigorously evaluate this capability, we simulate a large-scale annotation task on both the COCO-20$^i$ and LVIS-92$^i$ validation sets, mimicking a realistic deployment scenario in which new concepts must be annotated on the fly. For both datasets, we designate fold 0 as the test split, containing categories that are held out during training. Models are trained on the remaining folds. On COCO-20$^i$, we randomly sample 20 reference images, one per category in fold 0, and use them to guide the segmentation of 10,000 target images sampled from the same fold. On LVIS-92$^i$, which contains a larger and more fine-grained label space, we similarly sample 92 reference classes from fold 0 and use these to segment 10,000 target images. In both cases, the reference and target sets are fixed across all methods to ensure a fair comparison. Moreover, for all the methods, reference image features are computed once and cached, avoiding repeated backbone forward passes at inference time.

Table 6: **Evaluation of large-scale annotation efficiency on COCO-20$^i$ and LVIS-92$^i$ datasets.** We report mIoU and wall-clock annotation time (minutes) required to segment 10,000 target images using reference examples from unseen categories in fold 0. Reference features are precomputed and reused for fair comparison. Experiments conducted on an NVIDIA RTX 4090. SANSA achieves the best trade-off between accuracy and speed, demonstrating strong generalization and efficiency.

| Method | Fold 0 | | Total Annotation Time ↓ |
|---|---|---|---|
| | COCO mIoU (%) ↑ | LVIS mIoU (%) ↑ | |
| VRP-SAM [62] | 54.8 | 34.9 | 38 min |
| SegIC [44] | 55.1 | 46.7 | 77 min |
| GF-SAM [83] | 59.7 | 38.4 | 58 min |
| SANSA (**ours**) | **61.2** | **55.4** | **16 min** |

We benchmark SANSA against recent state-of-the-art few-shot segmentation methods, including GF-SAM [83], SegIC [44], and VRP-SAM [62], by measuring the total wall-clock time required to

segment all 10,000 images and evaluating segmentation quality via mean Intersection-over-Union (mIoU) against ground truth masks. Results are summarized in Tab. 6.

SANSA achieves the best trade-off between speed and accuracy across both COCO and LVIS. In particular, it completes annotation 2–5× faster than competing approaches while maintaining the highest mIoU. On LVIS, where the label space is significantly more complex, the gains are more pronounced: SANSA outperforms all baselines by a large margin (+8.7 mIoU over the best competitor), highlighting its strong generalization ability and annotation efficiency.

# D   Baselines and Ablation Studies

## D.1   Evaluating the Impact of SAM2 on Prior SAM-Based Pipelines

We analyze the effect of replacing SAM with SAM2 in existing Segment Anything-based methods, namely Matcher [42], GF-SAM [83], and VRP-SAM [62]. These methods follow a modular two-stage design: an external backbone (*e.g.*, DINOv2 for Matcher and GF-SAM, ResNet-50 for VRP-SAM) is used to perform feature matching, and the matched features are then used to generate geometric prompts that are passed to SAM, which operates at the image level to predict the final segmentation mask. We re-evaluate these approaches by replacing SAM with SAM2 in the final segmentation step, while leaving the matching backbone unchanged. Note that all these methods adopt the largest version of SAM (SAM-H), hence we replace it with the largest SAM2 version (SAM2-L). As shown in Tab. 7, the performance differences are negligible. This is consistent with findings from the SAM2 paper [54], which shows that SAM2-L yields only little gains over SAM-H in **image-level** promptable segmentation tasks. In other words, given the same prompts the segmentation quality is essentially unchanged, as both SAM and SAM2 excel at converting prompt geometry into accurate masks.

Table 7: **Performance comparison of Segment Anything-based methods replacing SAM with SAM2.** These methods prompt SAM at image-level, hence SAM2 additional capabilities (*e.g.* mask propagation) play no role in this setting. As a result, SAM2 provides only minimal gains for existing two-stage methods (Matcher [42], GF-SAM [83], VRP-SAM [62]).

| Method | Feature Matching Backbone | Segmentation Model | mIoU (%) |
|---|---|---|---|
| PerSAM [87] | SAM | | 23.0 |
| | SAM2 | | 23.6 |
| Matcher [42] | DINOv2 | SAM | 52.7 |
| | | SAM2 | 52.4 |
| VRP-SAM [62] | ResNet50 | SAM | 53.9 |
| | | SAM2 | 54.3 |
| GF-SAM [83] | DINOv2 | SAM | 58.7 |
| | | SAM2 | 58.9 |
| SANSA (**ours**) | SAM2 | | **60.2** |

These findings highlight the fundamental difference between SANSA and previous SAM-based approaches. While prior methods rely on external backbones for matching and treat SAM as a image-level decoder, our architecture tightly integrates all components into a unified pipeline. By leveraging SAM2 not only for mask prediction but also for feature matching and prompt encoding, SANSA ensures more effective feature reuse, leading to stronger performance and improved computational efficiency.

## D.2   Experiments with Different Backbones

In this section, we evaluate the performance of different SAM2 Encoder variants at varying scales. Table 8 presents a comparison of model parameters, inference speed (Frame Per Second), and mIoU across three versions of the model (Tiny, Base, and Large), with Frame Per Second (img/s) values computed on an NVIDIA RTX 4090.

Table 8: **Comparison of model parameters, inference speed (FPS), and mIoU** for different scales of the SAM2 encoder (Tiny, Base, Large) using Hiera. FPS are measured on an NVIDIA RTX 4090.

| SANSA Encoder | Total Parameters | Trainable Parameters | Frames Per Second (img/s) | mIoU (%) |
|---|---|---|---|---|
| Tiny | 40 M | 1.3 M | 50 | 51.1 |
| Base | 86 M | 3.3 M | 31 | 55.4 |
| Large | 234 M | 10 M | 15 | **60.2** |

All variants of SANSA offer a favorable trade-off between efficiency and accuracy, with consistent gains in mIoU as model capacity increases. Notably, the Large model achieves the highest mIoU of 60.2%, while still maintaining a practical inference speed of 15 FPS.

### D.3 Effect of the Training Objective

We next study the impact of our sequential conditioning objective. When $J = 1$, training reduces to the *standard few-shot segmentation objective*, where each target is predicted only from the reference image, without propagation. In contrast, we propagate predictions across multiple *unlabeled* targets ($J > 1$), encouraging the model to extract semantic structure that generalizes beyond individual pairs.

Table 9 shows that sequential conditioning consistently improves performance over the baseline with $J = 1$. We observe the largest gains with $J = 3$–$4$, after which results plateau.

Table 9: **Ablation on training sequence length** $J$. Using $J > 1$ improves over the standard few-shot objective ($J = 1$). Gains saturate at $J = 3$–$4$, which we use as default.

| $J$ | COCO-20$^i$ (1-shot) | LVIS-92$^i$ (1-shot) |
|---|---|---|
| 1 (no propagation) | 57.0 | 44.7 |
| 2 | 58.9 | 47.2 |
| 3 (**ours**) | **60.2** | **49.6** |
| 4 | 59.8 | 49.6 |
| 5 | 59.5 | 49.8 |
| 7 | 60.0 | 49.4 |

At inference, we emphasize that sequential conditioning is not used: each target is segmented independently given the annotated reference(s), following the standard few-shot segmentation protocol.

## E Additional Datasets and Qualitative Comparison

### E.1 Qualitative Comparison with Prior Methods

Fig. 10 showcases the performance of SANSA compared to GF-SAM [83] and SegIC [44] on samples from the LVIS-92$^i$ benchmark. Each row visualizes a reference image with an annotated mask and the corresponding target image. We report predictions from the best-performing baselines (GF-SAM [83] and SegIC [44]), our approach (SANSA), and the ground truth segmentation. For clarity, the name of the queried object class is also provided (note, however, that none of the models have access to the class label during inference).

These examples illustrate SANSA ability to resolve fine-grained distinctions and spatial part-awareness, addressing some of the core challenges of LVIS-92$^i$, where semantically close categories and partial object prompts are common. For instance, in the example where the reference depicts a *lamb*, SANSA is able to distinguish it from the semantically similar *sheep*. Similarly, it correctly segments a *gazelle* while avoiding confusion with adjacent *zebras*. In another case, SANSA successfully discriminates between a plush *monkey* and a visually similar plush *horse*, despite both being soft toys with overlapping colors and textures. Moreover, SANSA exhibits part-level understanding. When prompted with a reference showing a *t-shirt*, GF-SAM oversegments the full person, while SegIC narrowly restricts to clothing regions. In contrast, SANSA accurately segments the t-shirt alone.

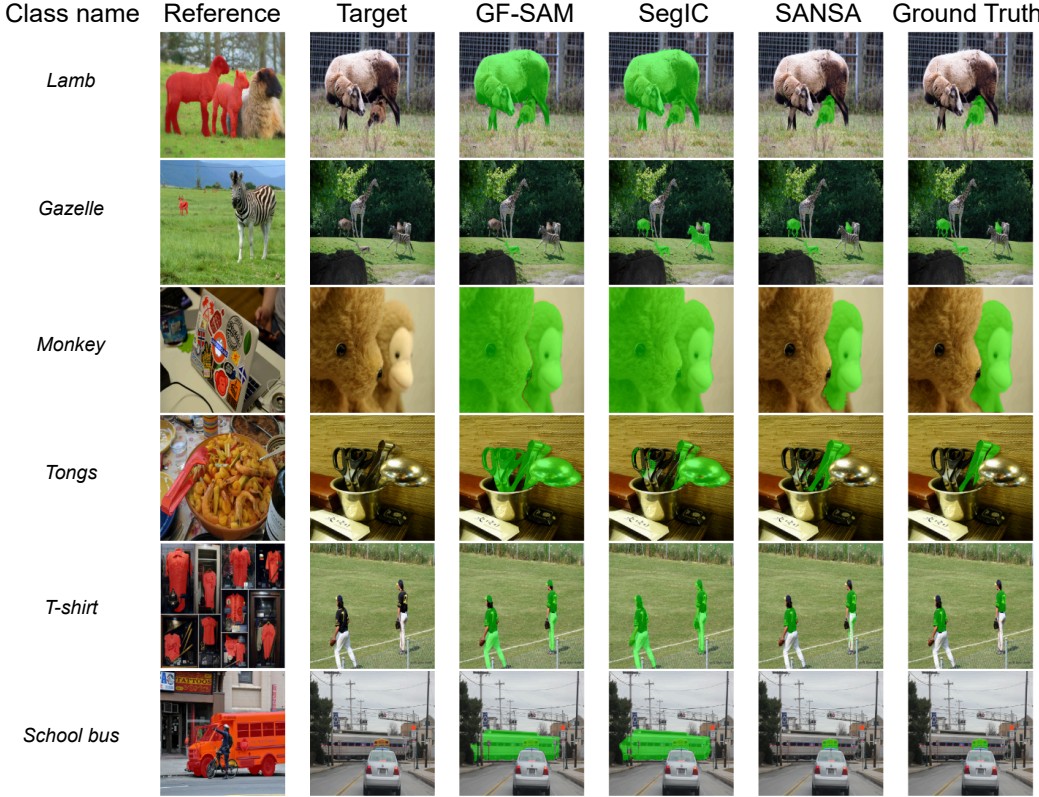

| Class name | Reference | Target | GF-SAM | SegIC | SANSA | Ground Truth |

Figure 10: **Qualitative comparison on the LVIS-92$^i$ benchmark.** For each example, we show: the annotated reference, the target image, predictions from GF-SAM and SegIC, our SANSA prediction, and the ground truth. The queried class name is shown for clarity, but is not used by any model.

These qualitative results complement our quantitative findings, further supporting our hypothesis that SANSA bridges high-level category understanding with fine-grained spatial precision.

### E.2 Additional Experiments: Pascal-5$^i$ and COCO → Pascal

While our main comparisons follow recent works [44, 89, 42] and focus on the benchmarks of COCO-20$^i$, LVIS-92$^i$, and FSS-1000, we additionally report results on Pascal-5$^i$ for completeness.

Table 10: **Evaluation on Pascal-5$^i$ and COCO→Pascal.** We report the mean Intersection-over-Union (mIoU) for each fold and the average across folds. All methods are evaluated in a strict few-shot setting, including both the standard Pascal-5$^i$ benchmark and the distribution-shift setting (COCO→Pascal), where models are trained on COCO-20$^i$ and tested on Pascal-5$^i$. $^†$ indicates training-free methods.

| Method | Pascal-5$^i$ | | | | | COCO→Pascal | | | | |
|---|---|---|---|---|---|---|---|---|---|---|
| | fold0 | fold1 | fold2 | fold3 | mean | fold0 | fold1 | fold2 | fold3 | mean |
| HSNet [46] [ICCV'21] | 67.3 | 72.3 | 62.0 | 63.1 | 66.2 | 47.0 | 65.2 | 67.1 | 77.1 | 64.1 |
| VAT [30] [ECCV'22] | 70.0 | 72.5 | 64.8 | 64.2 | 67.9 | 68.3 | 64.9 | 67.5 | 79.8 | 65.1 |
| AMFormer [72] [NIPS'23] | 71.3 | 76.7 | 70.7 | 63.9 | 70.7 | – | – | – | – | – |
| AENet [74] [ECCV'24] | 72.2 | 75.5 | 68.5 | 63.1 | 69.8 | – | – | – | – | – |
| HMNet [75] [NIPS'24] | 72.2 | 75.4 | 70.0 | 63.9 | 70.4 | – | – | – | – | – |
| PMNet [12] [WACV'24] | 71.3 | 72.4 | 66.9 | 61.9 | 68.1 | 71.0 | 72.3 | 66.6 | 63.8 | 68.4 |
| Matcher$^†$ [42] [ICLR'24] | 67.7 | 70.7 | 66.9 | 67.0 | 68.1 | 67.7 | 70.7 | 66.9 | 67.0 | 68.1 |
| GF-SAM$^†$ [83] [NIPS'24] | 71.1 | 75.7 | 69.2 | 73.3 | 72.1 | 71.1 | 75.7 | 69.2 | 73.3 | 72.1 |
| SANSA (**ours**) | 78.1 | 80.0 | 68.3 | 66.1 | **73.1** | 67.1 | 70.9 | 73.5 | 78.6 | **72.5** |

These experiments are conducted in the strict few-shot setting and include both the standard evaluation on Pascal-5$^i$, and the distribution-shift setting, COCO→Pascal, where models trained on COCO-20$^i$ are evaluated on Pascal-5$^i$ with folds repurposed to avoid any class overlap, as proposed originally in [8]. This setup aims to assess the robustness of models to shifts in data distribution, a condition often encountered in practical applications where training and deployment domains may differ. In Tab. 10, we compare with recent specialist few-shot methods (*e.g.*, HMNet [75], AENet [74]) and training-free methods employing two foundation models in cascade (*i.e.* Matcher [42], GF-SAM [83]). For the latter, we report identical results in both evaluation settings, as these models are frozen and evaluated directly on Pascal-5$^i$ without any training. The results confirm the competitive results of SANSA across benchmarks.

### E.3 Promptable Segmentation: Prompt Generation Process and Qualitative Results

We denote the type of annotation as $a_r^k \in \{\texttt{mask, point, box, scribble}\}$. Following [62, 90], point-based reference prompts are generated by randomly sampling between 1 and 20 points from the ground truth (GT). Scribble-based prompts are obtained, consistently with [62, 90], by using the free-form training mask generation algorithm proposed in [81], producing 1 to 20 scribbles per sample. Box-based prompts are derived by extracting object bounding boxes from the GT annotations.

In Fig. 11, we show the performance of SANSA in **one-shot promptable segmentation**. SANSA yields consistent and robust results across various prompts, including points, boxes, and scribbles.

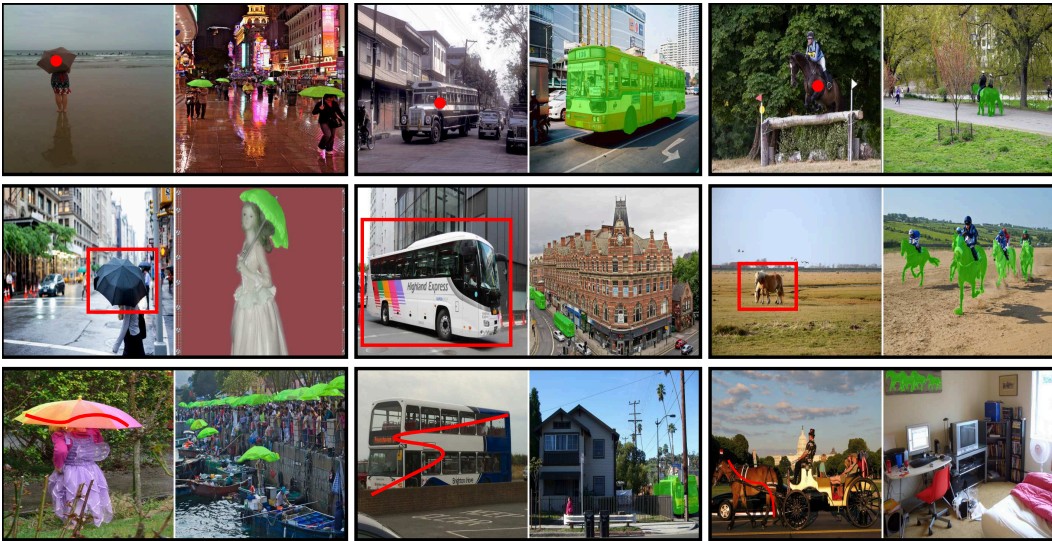

Figure 11: **Qualitative results for one-shot few-shot promptable segmentation.** The figure illustrates segmentation results using different types of prompts. The first row shows examples where the reference object is annotated with a point, the second row uses bounding box annotations, and the third row employs scribble-based annotations. Examples are extracted from COCO-20$^i$.

## F  Exploratory Analysis: Negative Prompts

Segment Anything 2 supports the use of *negative prompts* in interactive scenarios. These prompts, such as negative clicks on the image, allow users to correct mistakes by explicitly indicating regions that should not belong to the predicted mask. Since our method builds on SAM2, it is natural to ask whether this mechanism can be extended in the few-shot segmentation setting.

We conducted a small exploratory study to investigate this idea. We consider two ways of extending negative prompting:

- **Geometric negatives:** providing negative clicks on the target image at inference time, as in the interactive setting. This setup does not require any retraining and directly uses the mechanism already implemented in SAM2.

- **Semantic negatives:** providing additional examples of visually similar but semantically different categories alongside the reference image, to serve as contrastive context (*e.g.* including a photo of a horse to help disambiguate the target class zebra). We experimented with two simple variants:
  - *Training-free:* the negative reference is added to the `Memory Bank` with an *empty* mask.
  - *Training-based:* a small MLP maps the average feature of a negative object into a learnable negative token, similar to SAM2 handling of negative clicks.

The results in Table 11 indicate that both geometric and semantic negative prompts can provide measurable benefits in few-shot segmentation. However, this should be regarded as an exploratory direction: in particular, semantic negative prompting currently lacks standardized baselines and evaluation protocols, but represents a promising extension of the few-shot setting for future work.

Table 11: **Exploratory results with negative prompts on COCO-20$^i$.** Geometric negatives act as test-time corrections. Semantic negatives provide contrastive context.

| Method | mIoU |
|---|---|
| SANSA (baseline) | 60.2 |
| *Geometric Negative Prompts* | |
| +1 click | 60.8 |
| +3 clicks | 62.9 |
| +5 clicks | 64.5 |
| *Semantic Negative Prompts* | |
| Training-free | 61.2 |
| Training-based | 63.0 |

## F.1 Part Segmentation: Qualitative Results

In **one-shot part segmentation** (Fig. 12), SANSA demonstrates strong abilities to retrieve fine-grained object parts, highlighting the effectiveness of our method in learning part-level representations.

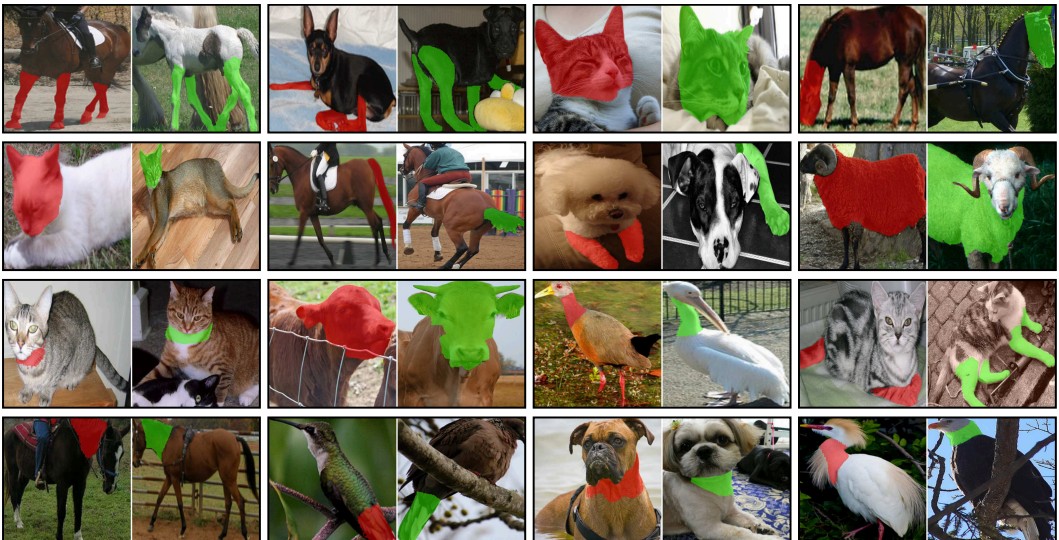

Figure 12: **Qualitative results for one-shot part segmentation.** Examples from Pascal-Part.

## G Exploring Dependence on Spatial Continuity in SAM2

The SAM2 model targets is tracking in temporally coherent videos. However, its architecture imposes no prior on spatial continuity. As detailed in Eq. (3), masks are propagated across frames through

a cross-attention between features, which allows all-to-all matching. Spatial coherence is thus not structurally enforced, but rather an emergent bias arising from pretraining on videos, encoded in the feature space. Our core finding is that this feature space contains latent semantic structure, entangled with spatial and appearance-related cues. Our adaptation disentangles these signals to isolate semantic information, enabling matching and segmentation across spatially independent inputs.

A natural question is how this implicit spatial continuity bias affects SAM2 behavior in practice. To investigate this, we adopt the spatial coherence metric of [86], which computes the average first-order gradient of the displacement field induced by feature matching. Low values indicate smooth, object-consistent correspondences, while high values reflect noisy or fragmented matches. We evaluate three conditions with progressively weaker spatial relationships across image pairs:

   (i) Consecutive video frames from DAVIS [51], where both temporal and spatial continuity are preserved.
  (ii) Flipped DAVIS frames, where the second image is mirrored, disrupting spatial alignment while preserving object identity.
 (iii) Independent images from TSS [64], with no spatial or temporal coherence.

As shown in Table 12, SAM2 performs well on natural videos as expected, but its coherence degrades when spatial alignment is perturbed and drops further on independent images. In contrast, SANSA maintains stable coherence across all scenarios, showing its ability to abstract semantic correspondences without relying on spatial continuity.

Table 12: **Spatial coherence across varying continuity conditions.** Lower is better.

| Setting | Continuity | | SAM2 ↓ | SANSA ↓ |
| --- | --- | --- | --- | --- |
| | Appearance | Spatial | | |
| (i) Video frames (DAVIS) | ✓ | ✓ | 1.5 | 1.7 |
| (ii) Flipped frames (DAVIS) | ✓ | ✗ | 5.5 | 2.1 |
| (iii) Independent images (TSS) | ✗ | ✗ | 9.1 | 4.1 |

# H  Implementation and Reproducibility

## H.1  Implementation details

We provide full implementation details to support reproducibility.

We adopt SANSA with Hiera-Large [58] as the visual encoder. Hiera is a hierarchical Transformer with channel dimensions [144, 288, 576, 1152]. AdaptFormer [14] adapters are inserted into the last two hierarchical blocks of the encoder (layers 9–48), corresponding to channel dimensions 1152 and 576. In the **strict few-shot setting**, the adapter hidden size is set to $0.3\times$ the block channel dimension, yielding hidden sizes of 346 (for 1152) and 173 (for 576). This configuration introduces about 10M trainable parameters, while all SAM2 weights remain frozen. In the **generalist setting**, the adapter hidden size is increased to $0.8\times$ the block channel dimension, which expands the adapters and raises the total number of trainable parameters to about 25M. Training in both settings is performed with AdamW (learning rate $10^{-4}$) and gradient clipping to 1.0. We train for 5 epochs in the strict setting and 20 epochs in the generalist setting, with a training sequence length of $J{=}3$. Supervision combines Binary Cross Entropy loss and Dice loss, equally weighted (1.0), applied to the predicted segmentation mask and ground truth. We train with batch size 32 on 8 A100 GPUs.

For clarity, all relevant training and architecture details are summarized in Tab. 13.

## H.2  Clarification of Few-Shot Segmentation Evaluation Settings

To avoid ambiguity in terminology and evaluation protocols, we first clarify the conventions followed in our work. In traditional few-shot segmentation literature [38, 69, 36, 41, 22], the annotated image is typically referred to as the *support* image, while the unannotated image to be segmented is called the *query* image. However, more recent works [62, 42, 83] have shifted terminology, denoting the

Table 13: **Summary of architecture and training hyperparameters.**

| Component | Value / Description |
|---|---|
| Base model | SAM2 (frozen) |
| Visual encoder | Hiera-Large [58] |
| Adapter type | AdaptFormer [14] |
| Adapted layers | Last two blocks (layers 9–48) |
| Adapter hidden size | $0.3\times$ (strict) / $0.8\times$ (generalist) of block dim. |
| Trainable parameters | $\sim$10M (strict) / $\sim$20M (generalist) |
| Optimizer | Adam |
| Learning rate | $10^{-4}$ |
| Gradient clipping | 1.0 |
| Batch size | 32 |
| Training epochs | 5 (strict) / 20 (generalist) |
| Training sequence length | $J = 3$ |
| Loss function | BCE (1.0) + Dice (1.0) |

annotated image as the *reference*, and the image to be segmented as the *target*. We adopt this updated nomenclature throughout our work.

Beyond terminology, few-shot segmentation is commonly evaluated under two distinct settings: the strict few-shot setting and the more recent generalist in-context setting. We describe both below to clarify the assumptions and evaluation scope of our method.

**Strict Few-Shot Segmentation.** Early methods in few-shot segmentation [38, 69, 36, 41, 22, 68, 84, 47, 46, 30] adopt a meta-learning framework, where the goal is to segment novel object classes given only a few annotated examples. Training proceeds episodically: each episode samples a subset of classes from a predefined training set, along with labeled reference images and corresponding unlabeled target images. The model must then predict segmentation masks for target images that contain instances of the same classes shown in the support set. A key assumption of this setup is the strict separation between training (seen) and evaluation (unseen) classes. This constraint ensures that performance reflects a model ability to generalize to entirely novel categories. To emphasize this evaluation protocol, the setting has recently been referred to [89, 44] as *strict* few-shot segmentation.

**Generalist in-context Setting.** The notion of in-context learning was popularized in natural language processing by GPT-3 [9], where models perform new tasks by conditioning on a sequence of input-output examples, without parameter updates. Rather than being explicitly trained for a single task, these models learn to infer the intended behavior directly from the prompt structure, enabling flexible adaptation to a wide range of downstream tasks. Inspired by this idea, recent vision models such as SegGPT [71] and Painter [70] reformulate segmentation as a conditional image generation problem. These models cast segmentation as a type of image-to-image translation or inpainting: prompts consist of paired inputs and output masks provided as images, and the model completes the target segmentation given these visual exemplars. This formulation allows generalization across a broad set of tasks, such as semantic, instance, and part segmentation, without retraining.

Within few-shot segmentation literature, Matcher [42] builds on this paradigm and proposes the *in-context setting*, where the model is prompted with examples and conditioned to produce the desired output. This formulation enables a single generalist model to address a wide spectrum of segmentation tasks, including few-shot segmentation, few-shot part segmentation, and video object segmentation. Several recent methods, including DiffewS [89], SegIC [44], and our SANSA, adopt this generalist perspective. These approaches operate in a more flexible evaluation regime, where prompts may include both seen and unseen classes, relaxing the disjoint class constraint of the strict setting.

### H.3   Datasets

For strict few-shot segmentation, separate models are trained on COCO-$20^i$, LVIS-$92^i$ and FSS-1000 following their respective $k$-fold splits or fixed partitions. In contrast, the generalist in-context setting

involves training a single model under three progressively more comprehensive configurations: a minimal setup with ADE20K and COCO; an extended setup including LVIS following SegIC [44]; and a further extended configuration incorporating PACO, following [71, 40], to mitigate object-level bias and improve part segmentation. Below, we provide brief descriptions of each dataset used in our experiments.

**COCO-20$^i$** [48] is constructed from the MSCOCO dataset [39], containing 80 object categories split into four disjoint folds. Each fold includes 60 training classes and 20 test classes. This dataset provides a challenging setting due to its large-scale nature and diverse object appearances.

**LVIS-92$^i$**, introduced by Matcher [42], is derived from the LVIS dataset [24] and presents a more challenging few-shot segmentation benchmark, emphasizing long-tail distributions. It consists of 920 classes that appear in at least two images, split into ten folds for cross-validation. This dataset introduces significant variations in object appearances and class distributions.

**FSS-1000** [23] is a large-scale few-shot segmentation dataset containing 1,000 classes with pixel-wise annotations, divided into 520 training, 240 validation, and 240 test classes. Unlike LVIS-92$^i$ and COCO-20$^i$, which use folds, FSS-1000 follows a fixed class partitioning.

**PASCAL-Part**, introduced by Matcher [42], is based on PASCAL VOC 2010 [21] and its part annotations released in [15] and provides fine-grained object part annotations. It consists of four superclasses: animals, indoor objects, persons, and vehicles. The dataset contains 56 object parts across 15 categories. Given its focus on object parts rather than whole objects, this dataset presents a challenging segmentation task.

**PACO-Part**, is derived from PACO, which provides part annotations for 75 object categories. Matcher [42] proposes a k-fold split of the 456 object part classes, from which 303 classes with at least two samples are retained. The dataset is split into four folds, each containing 76 object parts. Similar to PASCAL-Part, PACO-Part is designed for evaluating one-shot part segmentation models, but it includes a larger and more diverse set of categories.

**ADE20K** [88] is a large-scale semantic segmentation dataset containing 20,210 images annotated with 150 semantic categories at the pixel level. It includes a diverse range of indoor and outdoor scenes, covering natural landscapes, urban environments, and everyday objects.

