# OpenReview forum: "SANSA: Unleashing the Hidden Semantics in SAM2 for Few-Shot Segmentation"
_NeurIPS.cc/2025/Conference — NeurIPS 2025 spotlight_

### Official Review · Reviewer_Cxfs · 2025-06-27

**Clarity:** 4
**Significance:** 3
**Originality:** 3
**Rating:** 5
**Confidence:** 4

**Summary:**

This paper repurposes the SAM2 object tracking framework for the Few-Shot Segmentation (FSS) task. FSS requires the model to segment objects in a query image based on feature matching with a given support image. While SAM2 is inherently capable of instance-level object matching and tracking, it lacks the semantic-level generalization needed for FSS. To address this, the authors introduce an adaptor module based on AdaptFormer, which enables the model to adapt to class-level segmentation tasks. The training objective is designed around support and query sets, and predicts the mask conditioned on its previous prediction. This setup enables more coherent and context-aware segmentation results.

**Questions:**

Please refer to the Weaknesses part.

**Ethical Concerns:**

["NO or VERY MINOR ethics concerns only"]

**Final Justification:**

The authors have successfully addressed all of my concerns. While the application is limited to few-shot segmentation, the paper presents a clear motivation that is well aligned with the proposed method. Overall, I believe this is a solid and well-executed work. Therefore I raise my score to 5.

**Limitations:**

Yes

**Quality:**

4

**Strengths And Weaknesses:**

**Strengths**
- The paper is clearly written and easy to follow.
- The motivation for adapting the SAM2 framework to Few-Shot Segmentation (FSS) is well presented, with an analysis of SAM2’s capabilities. Also, exploiting SAM2 enables segmentation from diverse input prompts.
- By using an adaptor, the method preserves the original object tracking performance of SAM2 while extending it effectively to FSS.
- The proposed method outperforms prior approaches in both accuracy and efficiency. Design choices are well supported through ablation studies.

**Weaknesses**
- Figure 1 lacks an explanation for the qualitative results shown in the bottom part. It is unclear whether these are SAM2's zero-shot outputs or ground-truth segmentations. If they are SAM2 zero-shot results, the rightmost example may be an inappropriate illustration, as performance degradation is expected in such cases.
- In Figure 2, it is not explained whether the colors carry semantic meaning. If they do, adding a colorbar would improve clarity. For instance, in the right example, SAM2's output does not appear significantly worse, so clarification would help contextualize the comparison.
- The paper lacks details regarding the training objective part. It is unclear what specific values or configurations were used for hyperparameters such as r and J. Do these values vary depending on whether the model is trained under the 1-shot or 5-shot setting?
- The construction of the training image set is only briefly mentioned. Was the sampling purely random based on class labels, or were there additional criteria? And, is the order of images within each training set shuffled during training? Given that object size and the number of objects can vary significantly across images, this might impact learning stability and generalization.
- Relatedly, since the model conditions each query prediction on previous predictions, it is unclear how robust the method is to poor initial masks. If the initial prediction is inaccurate, is there a risk of cascading errors or convergence to suboptimal solutions (e.g., local minima)? An analysis or discussion of such failure cases would strengthen the paper.

---

> ### Author Rebuttal · Authors · 2025-07-29
>
> We sincerely thank the Reviewer for the positive and constructive feedback. We are encouraged to see appreciation for the **clarity** of the paper, the **motivation** behind adapting SAM2 to the few-shot segmentation setting, the **flexibility** to support diverse prompts, and the strong **results** achieved in both **performance and efficiency**.
> Below, we address the remaining concerns in detail, providing clarifications and additional analyses. We hope these additions further strengthen the paper and improve its presentation.
>
> ----------
> 1.
>
> > **W1:** *Unclear whether qualitative results shown in the bottom of Figure 1 are SAM2 zero-shot outputs or ground-truth segmentations.*
>
> We appreciate the Reviewer observation and clarify that the bottom part of Figure 1 shows **ground-truth masks**. These examples are meant to visually contextualize the two types of scenarios discussed in our toy experiment: *low semantic shift / high visual similarity* (e.g., medical scans, left) and *high semantic shift / low visual similarity* (e.g., cruise ship vs. rowboat, right). The red masks simply indicate the object of interest. We will revise the caption accordingly.
>
> More broadly, the right part of the figure motivates our core research question: *Can the semantic capabilities observed in other vision foundation models (VFMs) also be found in SAM2?* A growing line of research [21, 37, 67, 69, 70] has demonstrated that semantic structure is encoded in the features of **frozen** VFMs such as DINOv2 and Stable Diffusion. We conducted this toy experiment to **test** whether **similar semantic properties** could be observed in the frozen SAM2. The aim of Figure 1 is thus to set the stage for the rest of the paper: to highlight that, **unlike in other VFMs, semantic structure in SAM2 does not naturally emerge in zero-shot**.
>
> ------------
> 2.
>
> > **W2:** *In Figure 2 (PCA), it is not explained whether the colors carry semantic meaning.*
>
> We thank the Reviewer for the helpful observation and agree that clarifying the caption would improve interpretability.
> In Figure 2 we apply Principal Component Analysis (PCA) to the extracted feature vectors and map the first three principal components (PC1, PC2, PC3) to the red, green, and blue channels, respectively. Consequently, each point RGB color encodes its position in the 3D PCA-projected space. This approach is widely used to **analyze the global structure** of high-dimensional embeddings [43, 54, 70].
> Importantly, the colors do *not* correspond to ground-truth class labels and should not be interpreted categorically. Rather, they reflect spatial proximity in the projected space: points with similar colors lie close in the 3D PCA space. This allows us to **qualitatively assess what types of variation are captured by the leading axes**. In the case of frozen SAM2, although some hue variation exists (*e.g.*, surfboards appear slightly violet and people greenish-blue), the overall color distribution lacks clear separation across the four object categories (*dog, frisbee, person, surfboard*). This suggests that the principal components capture a mixture of low-level or instance-specific signals, rather than pure semantics. In contrast, **SANSA** yields clearly distinct and consistent colors (*e.g.*, orange, yellow, violet, light blue), indicating that **semantic structure dominates the main axes of variability.**
>
> A more comprehensive evaluation is reported in **Figure 4a**, where we use a 2D PCA projection with ground-truth color labels to visualize **class-level clusters**.
>
> ---
> 3.
>
> > **W3:** *Details regarding the training: specific values for hyperparameters such as r and J. Do these values depends on whether the model is trained under the 1-shot or 5-shot setting?*
>
> We thank the reviewer for the comment and are happy to clarify. We use a *training sequence* length of **J=3**, meaning each episode consists of one annotated reference followed by three unlabeled target image. The number of annotated references during *training* is fixed to **r=1**. At training, these values **do not change**. At *inference time*, we simply provide either one (r=1) or five (r=5) annotated examples based on the benchmark protocol (as in Table 1 of main paper). This unified training strategy enables **seamless support for variable r-shot inference** (*e.g.*, 1-shot or 5-shot), without retraining or architectural modifications.
>
> Details on the training configuration, including the fixed value J=3, are included in **Section §G.1 Implementation Details of the Supplementary Material**. We appreciate the suggestion and will revise the text to also clearly state that r=1 is used during training.
>
> ----------------------------------------------
>
> 4.
>
> > **W5:** *If the initial prediction is inaccurate, is there a risk of cascading errors or convergence to suboptimal solutions (e.g., local minima)?*
>
> We appreciate the reviewer thoughtful question. The sequential conditioning mechanism is used **exclusively during training to promote robust and generalizable representations**. By requiring the model to propagate semantic information across multiple unlabeled targets, this objective discourages overfitting to individual image pairs and encourages the emergence of shared, semantic structure, **forcing the network to disentangle high-level concepts from low-level visual cues**.
>
>
> To directly address the concern about potential cascading errors or instability from inaccurate early predictions, we perform an ablation on the training sequence length J, which defines how many unlabeled target frames are segmented after the annotated reference. When J=1, no sequential conditioning or mask propagation occurs. If our formulation were prone to cascading errors, we would expect performance to degrade for larger J. Instead, we observe that all settings with J>1 consistently outperform the baseline with J=1, **demonstrating the stability and benefits of our training strategy**.
>
> **Table R5: Ablation on Mask Propagation**, † Newly added for rebuttal.
> | \( J \) | COCO-20i (1-shot) | LVIS-92i (1-shot) |
> |--------|-------------------|-------------------|
> | 1 (no mask propagation)      | 57.0             | 44.7             |
> | 2      | 58.9 †            | 47.2 †            |
> | 3      | **60.1**         | **49.8**        |
> | 4      | 59.8 †            | 49.6 †            |
> | 5      | 59.5 †            | 49.8 †            |
> | 7      | 60.0 †            | 49.4 †            |
> | 10      | 59.9 †            | 49.5 †            |
>
> These results provide strong empirical evidence that our sequential training formulation is **not vulnerable to cascading errors**, but instead enhances the quality and generality of the learned features.
>
> Lastly, we clarify that this sequential conditioning mechanism is used only during training. At **inference time**, our method follows the standard few-shot segmentation protocol: each target image is segmented independently, using only the annotated reference(s), without mask propagation or reliance on previous predictions.
> We thank the reviewer for this insightful observation and will incorporate the ablation and its analysis in the revised manuscript.
>
>
> -------------------------
>
> 5.
>
> > **W4:** *The construction of the training image set is only briefly mentioned.*
>
> We appreciate the Reviewer suggestion to clarify the training data construction process. We followed the **standard protocol adopted in few-shot segmentation literature** [16, 17, 20, 39, 40]. For each training episode, a class is randomly selected from the training set, and images containing at least one instance of that class are sampled uniformly at random. No filtering is applied based on object size, instance count, or other properties. The order of images within episodes is shuffled during training.
> Since no special constraints are introduced, we did not emphasize this aspect in the main paper. However, we agree that adding this clarification can improve completeness, and will revise the manuscript accordingly.
>
> ---
>
> We once again thank the Reviewer for the time dedicated to evaluating our work and for the constructive questions and observations raised. We hope our responses have clarified all points and helped improve the overall quality of the paper.

---

> > ### Comment · Reviewer_Cxfs · 2025-08-05
> >
> > Thank you for your responses.
> >
> > The clarification for Q1 helped me better understand Figure 1. Given this, it might be beneficial to include a similar plot that shows the performance gap between your method (SANSA) and state-of-the-art methods, e.g., Δ(SANSA zero-shot − SOTA): mIoU. This kind of visualization could more clearly emphasize the strength of the proposed method.
> >
> > The responses to Q2 through Q5 were also helpful for my understanding, and I hope these clarifications will be incorporated into a future revision.

---

> > > ### Author Response · Authors · 2025-08-05
> > >
> > > We sincerely thank the Reviewer for the positive and constructive feedback. We're very happy that the clarifications addressed the concerns raised, and we truly appreciate the time and care dedicated to reviewing our work.
> > > The comments raised were very helpful, and we will make sure to reflect them in the final version to further strengthen the clarity and completeness of the paper.

---

### Official Review · Reviewer_eWJd · 2025-07-03

**Clarity:** 3
**Significance:** 2
**Originality:** 2
**Rating:** 5
**Confidence:** 4

**Summary:**

This paper proposes SANSA, a method that adapts the frozen SAM2 model for few-shot segmentation by inserting AdaptFormer blocks into the last two layers of SAM2's image encoder. The authors argue that SAM2 features already contain semantic information that is entangled with low-level signals, and their adaptation method disentangles this to make semantic content explicit. The approach achieves strong performance on multiple benchmarks while preserving SAM2's original capabilities and requiring only 10M additional trainable parameters.

**Questions:**

During the rebuttal stage, please answer the following question: I could raise my score if the authors addressed my concerns.
1. Could you provide systematic ablation studies comparing different adapter placement strategies (e.g., early layers, middle layers, all layers) to justify the "last two layers" choice?
2. What insights can you offer about why AdaptFormer works explicitly better than other adaptation methods for disentangling semantic information in SAM2?
3. How does your method fundamentally differ from standard fine-tuning approaches beyond the efficiency perspective?
4. Could you demonstrate that the semantic disentanglement is specific to your approach rather than a general property of any adaptation of SAM2 features?

**Ethical Concerns:**

["NO or VERY MINOR ethics concerns only"]

**Final Justification:**

All my concerns were solved after reading the reviewers' replies. I'm grateful for the authors' reclaim, and I understand the paper better now, so I will raise my rating to 4. After reading the final remarks, I finally raised my rating to 5 and appreciate the quality of this paper.

**Limitations:**

Yes, the authors acknowledge limitations in Section A.1, noting that their method inherits the design assumptions and limitations of SAM2.

**Quality:**

3

**Strengths And Weaknesses:**

Strengths:
1. The performance of the proposed method is quite good. It achieves state-of-the-art results on COCO-20i, LVIS-92i, and FSS-1000 with significant efficiency gains.
2. The authors provide a thorough investigation of the semantic structure through PCA decomposition and linear probing experiments.
3. This paper also provides detailed feature visualization and analysis, which clarifies the overall contribution of the paper.

Weaknesses:
1. Novelty is rather low. The core contribution is essentially applying AdaptFormer (an existing technique) to SAM2 (an existing model), which is a relatively straightforward engineering solution without significant algorithmic novelty. The author should more clearly claim their main contributions.
2. The manuscript occasionally suffers from dense prose, unclear transitions (e.g., Sections 3.1 and 3.2), and long paragraphs. This makes it challenging for readers less familiar with SAM2 or FSS to follow.
3. The paper lacks thorough ablation studies on adapter placement strategies, architectures, or hyperparameters. The choice of "last two layers" appears somewhat arbitrary without systematic justification. Also, there is limited theoretical insight into why this specific adaptation strategy works or how it fundamentally differs from other fine-tuning approaches beyond empirical observations

---

> ### Author Rebuttal · Authors · 2025-07-29
>
> We thank the Reviewer for the constructive feedback. We are glad to see that the **strong performance**, **efficiency gains**, and our **in-depth analysis**, including PCA decomposition, linear probing, and feature visualizations, were appreciated. In the following, we address the concerns raised with new experiments, expanded theoretical insights, and a clearer articulation of our contributions. We hope these additions clarify the novelty and significance of our work.
>
> ---
> 1.
> > **W1:** *The core contribution is essentially applying AdaptFormer to SAM2: the authors should more clearly claim their main contributions.*
>
> We thank the Reviewer for the opportunity to clarify the core contributions of our work. The novelty lies in a key insight: SAM2, though trained for class-agnostic object tracking, encodes a latent semantic structure (L51-L56) that can be disentangled and repurposed for few-shot segmentation (FSS).
> - From a **scientific** standpoint, we are the **first** to **identify**, **analyze**, and **validate** the presence of **semantic structure in SAM2 features**. We show in Supp. §C.2 that a simple linear probe on frozen features recovers class-level information, providing evidence that semantics are already embedded and do **not need to be learned**. Based on this, we propose a new perspective on few-shot segmentation: instead of relying on separate models (DINOv2 for **matching** and SAM for **segmentation**), we show that SAM2 alone can perform **both**, enabling a unified and streamlined solution. More broadly, this **repositions SAM2** as a standalone foundation for a wider class of pixel-level semantic tasks.
> - From a **technical** standpoint, we *i)* repurpose SAM2 memory mechanism by feeding it pseudo-videos, reinterpreting the temporal dimension as a collection of semantically related images. We *ii)* introduce a novel training protocol that promotes concept propagation across *multiple* unlabeled frames, encouraging the emergence of robust semantic correspondence. We *iii)* adapt the feature space to shift from low-level visual similarity to class-level alignment. Notably, the **architectural simplicity of our design is a consequence of our hypothesis**: if semantic structure had to be introduced through architectural modifications, it would undermine our claim that such structure already exists in SAM2.
> - From a **practical** standpoint, SANSA achieves SOTA results across FSS benchmarks, in-context learning scenarios, and part segmentation tasks. It does so while being 3× faster and up to 5× smaller than prior methods. Crucially, our model supports low-effort prompts (*e.g.*, points), making it well-suited for real-world applications such as data annotation.
>
> We will revise the manuscript to better frame and position our contributions.
>
> ---
> 2.
> > **Q2:** *Why does AdaptFormer work explicitly better than other adaptation methods? **Q4:** Is the semantic disentanglement specific to your approach rather or a general property of any adaptation of SAM2 features?*
>
> Building on the previous point, our results are **not meant to imply that semantic disentanglement arises specifically because of AdaptFormer**.
> We emphasize that our goal is **not** to engineer a dedicated adapter (L56-L60), but to investigate the representational structure in SAM2: **if semantic disentanglement depended on the design of the adapter, it would suggest that the semantics are induced by adaptation,** rather than being already embedded in the pretrained model.
>
> In this context, we detail what makes an adaptation strategy effective, expanding on results reported in the paper with additional experiments in Tab. R3.
> - As shown in Tab. 4 (main paper), basic adaptation strategies (*i.e.* LoRA, Adapter, AdaptFormer) yield similarly **strong improvements** of +25% w.r.t. frozen SAM2 (32.2%).
> - Importantly, LoRA is a linear adapter. The slight gains (~2%) observed with Adapter and AdaptFormer, both of which introduce a single ReLU non-linearity, suggest that simple **non-linear transformations** can **help** refine this structure, but do **not fundamentally reshape it**.
> - In Tab. 5, we study the effect of **adapter capacity**. Reducing the bottleneck dimension from 768 → 576 → 384 leads to progressively better generalization (58.2% → 59.6% → 60.1%). This indicates that **smaller adapters** are **better** at capturing the relevant structure in the pretrained features, **avoiding overfitting** to spurious correlations.
> - To further test this hypothesis, we **introduce a new experiment** with a recent SOTA adapter, **MONA** [A], which include depthwise convolutions and LayerNorm to increase functional complexity. Despite this, MONA underperforms (56.9%), suggesting that added complexity leads to overfitting rather than better representations.
>
> In summary, **semantic disentanglement is not unique to AdaptFormer, but also does not arise from arbitrary adaptation**. Our results suggest that effective adaptation requires **simplicity and constraint**: low-capacity, bottlenecked modules are best suited to uncovering SAM2 latent semantics. In contrast, more expressive adapters tend to overfit.
>
> **Table R3 Finetune vs Adaptation.** † Newly added for rebuttal
> |||Train Params (M)|mIoU (%)|
> |-|-|-|-|
> |**Frozen**|-|0|32.2|
> |**Finetuning (FT)**|Train from Scratch †|224|37.1|
> ||Full FT †|224|51.6|
> ||Decoder FT|4|52.1|
> ||QKV-only FT|50|55.3|
> ||Backbone FT|210|55.2|
> |**Adaptation**|LoRA|18|58.0|
> ||Adapter|20|59.4|
> ||Mona†|26|56.9|
> ||AdaptFormer (**SANSA**)|**10**|**60.1**|
> ||├─ Bottleneck: 768,576,384|26,16,10|58.2, 59.6, **60.1**|
>
> [A] "5%> 100%: Breaking performance shackles of full fine-tuning on visual recognition tasks." CVPR2025.
>
> ---
> 3.
> > **Q3:** *How does SANSA differ from standard fine-tuning approaches beyond the efficiency perspective?* **W3:** *Theoretical insight into why adaptation strategy works or how it differs from other fine-tuning?*
>
> Our choice to keep SAM2 frozen is **not motivated by efficiency**, but by the need for **generalization**, which is the core challenge in few-shot segmentation. All recent state-of-the-art methods [35, 67, 72, 37] do *not* finetune  pretrained Vision Foundation Models to avoid overfitting when learning from limited data. SANSA follows this paradigm. As shown in the **Tab. 1 of the Supp.** (Sec. §C.3) and in **Tab. R3 above**, full or partial fine-tuning of SAM2 consistently results in worse generalization. We further added two new experiments (training from scratch and full fine-tuning), which confirm the same trend.
>
> From a **representation learning perspective**, SAM2 encodes rich, reusable features that capture both appearance and **latent semantics**. Lightweight bottleneck adapters such as AdaptFormer or LoRA apply **low-rank transformations** that reorient the embedding space toward task-relevant semantics while **preserving** the underlying pretrained structure. This can be viewed as a form of semantic disentanglement under a strong inductive prior, *i.e.* the pretrained model [B, C]. In contrast, directly fine-tuning the model parameters risks erasing these priors, particularly in few-shot or continual learning settings, where **data is limited and distributions shift at test time** [C, D].
> Our empirical findings (Tab. R3: *Finetune vs Adaptation*) align with this interpretation: adaptation successfully leverages and restructure existing structure, thus promoting generalization.
>
> [B] "Fine-tuning can cripple your foundation model; preserving features may be the solution." TMLR2024
> [C] "Fine-tuning can distort pretrained features and underperform out-of-distribution." ICLR2022
> [D] "Ranpac: Random projections and pre-trained models for continual learning." NeurIPS2023
>
> ---
> 4.
> >  **Q1, W3**: *The paper lacks ablations on adapter placement (to justify the "last two layers" choice), architectures, or hyperparameters.*
>
> We thank the reviewer for raising this important point. While prior works typically insert adapters throughout the **entire** network to maximize representational capacity [E,F,G,H], we purposefully limit adaptation to the last 2 hierarchical blocks of the backbone, which are known to encode **high-level semantics**. This design choice stems from **our core objective**: **disentangle semantic structure**. In response to the reviewer suggestion, we conducted a systematic ablation on adapter placement (Tab. R4) to better justify our design choice.
>
> **Table R4. Adapter Placement**
> |Placement|Stages|mIoU (%)|
> |-|-|-|
> |None|–|32.2|
> |All|0,1,2,3|59.4|
> |Early|0,1|38.7|
> |Middle|1,2|57.9|
> |Late|2,3|**60.1**|
>
> The ablation confirms that, for **semantic disentanglement**, it is beneficial to adapt the latest stages of the backbone.
>
> These results complement our ablations on the core design elements, including **Adapter architecture** (Tab. 4, main paper; expanded in Tab. R3), **Adapter capacity** (Tab. 5, main paper), **Adaptation vs. fine-tuning** (Tab. 1, Supp; expanded in Tab. R3). We hope the additional experiment fully addresses the reviewer concern.
>
> [E] "Side Adapter Network for Open-Vocabulary Semantic Segmentation." CVPR2023
> [F] "Exploring Temporally-Aware Features for Point Tracking." CVPR2025
> [G] "AdapterFusion: Non-Destructive Task Composition for Transfer Learning." EACL2021
> [H] "Vision Transformer Adapter for Dense Predictions." ICLR2023
>
> ---
> 5.
> > **W2:** *Dense prose; challenging for readers less familiar with SAM2 or FSS.*
>
> We sincerely thank the reviewer for this helpful observation. We recognize that clarity is essential, particularly when introducing concepts that may be unfamiliar to all readers. We will revise the manuscript in Sections 3.1 and 3.2 by smoothing transitions and simplifying technical descriptions where needed. While other reviewers found the presentation to be clear, we appreciate that individual perspectives vary, and we are committed to ensuring that the final version is accessible to a broader audience through careful revision.

---

> ### Comment · Reviewer_eWJd · 2025-08-01
>
> Thank you for the detailed rebuttal. I think such contributions to representation learning are sufficient after reading the all rebuttals. Therefore, I have increased my score to 4.

---

> > ### Author Response · Authors · 2025-08-01
> >
> > We are glad that our responses helped clarify the contributions and improve the overall quality of the paper.
> > We sincerely thank the Reviewer for the time dedicated to evaluating our work and for the constructive comments and questions.

---

### Official Review · Reviewer_6sqR · 2025-07-03

**Clarity:** 3
**Significance:** 3
**Originality:** 3
**Rating:** 5
**Confidence:** 4

**Summary:**

This paper proposes a framework to unleash the hidden semantic capabilities of SAM2 for few-shot segmentation. A lightweight adapter module is applied after the image encoder to reinterpret the temporal dimension as semantically related features. The results set a new state-of-the-art in strict few-shot segmentation across multiple benchmarks.

**Questions:**

Instead of feature adaptation from the frozen original SAM2 weights, why not retrain a few-shot learning with the whole weights using the proposed architecture?

**Ethical Concerns:**

["NO or VERY MINOR ethics concerns only"]

**Final Justification:**

The rebuttal addressed my concerns in Weakness about lacking theoretical analysis and insights, as well as in Questions about the comparison to the method training from scratch. Therefore I raised the score to 5.

**Limitations:**

Yes

**Paper Formatting Concerns:**

In Figure 2, it’s better to indicate PCA in the second row as well.

**Quality:**

3

**Strengths And Weaknesses:**

Strengths:
1.The paper is well written with clear method illustration, feature analysis, and result presentations.
2.The motivation to adapt original SAM2 for few-shot segmentation is interesting and technically sound. The paper shows a clear focus and boundary on few-shot learning with fair claims.
3.The experiment design is comprehensive with promising results.

Weaknesses:
The proposed framework is technically modified from SAM2. As SAM2 is designed for object tracking given videos with spatial relationships, there lacks a theoretical analysis and insights of the adaptation from continual frames to spatially independent pseudo-videos in the feature space. Therefore limits the methodological contribution is limited.

---

> ### Author Rebuttal · Authors · 2025-07-29
>
> We thank the Reviewer for the positive and encouraging feedback. We are glad that the **motivation** to adapt SAM2 to few-shot segmentation was found to be both **interesting** and **technically sound**, and the experiments to be **comprehensive**. We also appreciate the positive remarks on the **clarity of the analysis and presentation**. We hope that the additional insights and empirical evidence provided below further strengthen the paper and address the remaining concerns.
>
> ---
>
> 1.
> > **Q1:** *Instead of feature adaptation, why not retrain the model with the whole weights?*
>
> This is an interesting point, which directly relates to a fundamental aspect of few-shot segmentation (FSS). The core challenge in FSS is **generalization**: the ability to transfer to novel, unseen categories from only a handful of labeled examples. For this reason, recent state-of-the-art FSS methods [35, 37, 67, 72] rely on **frozen** Vision Foundation Models (VFMs) such as DINOv2 and Stable Diffusion. While these models provide rich, transferable features acquired through large-scale pretraining, it has been shown that in low-shot settings [A, B, C] **retraining or fine-tuning** leads to **degrading the pre-trained representations** and ultimately poor generalization.
>
> We explicitly analyze this point in **Section §C.3 of the Supplementary Material**, where we compare our *adaptation* with several *fine-tuning* strategies, including decoder-only, QKV-only, and full-backbone tuning. These strategies update between 4M and 210M parameters, yet all fall short of our proposed adaptation.
>
> In response to the reviewer suggestion, we additionally **include two further baselines**: *i)* full fine-tuning of the entire model, starting from SAM2 pretrained weights; and *ii)* training the SAM2 architecture from ImageNet initialization. Both configurations yield substantially lower performance. This **confirms the theoretical expectation** that, in few-shot regimes, *i)* full fine-tuning degrades the generalization acquired during pretraining due to overfitting on limited categories, while *ii)* training from a generic initialization lacks the semantic priors necessary to generalize to novel classes.
>
> **Table 1 from Section §C.3 of the Supplementary Material**, † Newly added for rebuttal
> | Adaptation Strategy           | Train Params | COCO-20i mIoU (%) |
> |-------------------------------|------------|-------------------|
> | Training from Scratch†        | 224M       | 37.1              |
> | Full Finetuning†   | 224M       | 51.6              |
> | Decoder Finetuning           | 4M         | 52.1              |
> | QKV-only Finetuning          | 50M        | 55.3              |
> | Full Backbone Finetuning     | 210M       | 55.2              |
> | **SANSA (Ours)**             | **10M**    | **60.1**          |
>
>
> [A] "Fine-tuning can cripple your foundation model; preserving features may be the solution." TMLR2024
> [B] "Fine-tuning can distort pretrained features and underperform out-of-distribution." ICLR2022
> [C] "Ranpac: Random projections and pre-trained models for continual learning." NeurIPS2023
>
> ---
>
> 2.
> > **W1**: As SAM2 is designed for object tracking given videos with spatial relationships, there lacks a theoretical analysis and insights of the adaptation from continual frames to spatially independent pseudo-videos in the feature space.
>
>
> We thank the Reviewer for the insightful comment, which raises an important point about the adaptation of SAM2 to spatially independent inputs in the few-shot segmentation scenario.
>
> Although SAM2 is designed for tracking in temporally coherent videos, its architecture does **not impose any prior on spatial continuity**. In fact, as detailed in Eq. (2), masks are propagated through the video based on a cross-attention between frame features, that allows all-to-all matching. Therefore, spatial coherence is not structurally enforced, but rather an emergent **bias** that **arises from pretraining** on videos, encoded in the **feature space**.
> Our core finding is identifying, within such feature space, **the presence of latent semantic structure** (*cf* Linear probing, Section §C.2 Supp) entangled with spatial and appearance-related information (2D PCA, Section 3.4 Main). Our adaptation disentangles these signals to isolate the semantic information, enabling matching and segmentation across **spatially independent** inputs.
>
> We agree that providing in-depth analyses and insights on the effect of this adaptation is important to ground our results. Indeed, **Section 3.4** of the main paper is devoted to study how this **adaptation reshapes the feature space**, by showing the principal component analysis of the feature space before and after the disentanglement. Please note that, due to space constraints, a more **thorough investigation** with additional insights and theoretical justifications is presented in the Supplementary (**Section  §C: Emergence of Semantic Representations**).
>
> While our analysis primarily focused on the **emergence of semantic representations**, we appreciate the Reviewer suggestion to further examine the **effect of adaptation** from **continual frames to spatially independent pseudo-videos** in the feature space. To this end, we have performed a new analysis adopting the **spatial coherence** metric from [70], which computes the average first-order gradient of the displacement field produced by matching: low values indicate smooth, object-consistent matches, while high values reflect noisy or fragmented correspondences. Through the lens of this metric, we investigate the extent to which SAM2 relies on spatial correlations in the data, and how our adaptation mitigates this dependence. We consider three settings, with progressively looser spatial relationships across image pairs:
>
> * i) **Video frames**: pairs of consecutive frames from DAVIS [D], where both temporal and spatial continuity are preserved, *e.g.*, a Labrador appears in the bottom left in the first frame and slightly to the right in the second.
> * ii) **Flipped frames**: same DAVIS frame pairs, but the second image is horizontally flipped, disrupting spatial alignment while preserving object identity, *e.g.* the Labrador appears in the same pose but mirrored, now on the opposite side.
> * iii) **Independent images**: pairs of unrelated images from the same category, sampled from TSS [E], with no spatial or temporal coherence, *e.g.* the first image shows a Labrador, and the second a Dalmatian.
>
> **Table R2 Spatial Coherence across varying continuity conditions**
> *(Lower is better)*
>
> | Setting                           | Appearance Continuity | Spatial Continuity | SAM2 ↓ | SANSA ↓ |
> |----------------------------------|:-------------------:|:------------------:|:------:|:-------:|
> | *i)* Video frames (DAVIS)         | ✓                   | ✓                  | **1.5**| 1.7     |
> | *ii)* Flipped frames (DAVIS)      | ✓                   | ✗                  | 5.5    | **2.1** |
> | *iii)* Independent images (TSS)   | ✗                   | ✗                  | 9.1    | **4.1** |
>
>
> Results show that SAM2, as expected, performs well under *i)*, degrades under *ii)*, and struggles under *iii)*, confirming that its coherence partially relies on a continuity in the video frames. In contrast, SANSA maintains coherence across all scenarios, showing its **ability to abstract semantic** correspondences without relying on spatial continuity.
>
> [D] "A benchmark dataset and evaluation methodology for video object segmentation." CVPR16
> [E] "Joint recovery of dense correspondence and cosegmentation in two images." CVPR16
>
> ---
>
>
>
> We sincerely thank the Reviewer for these insightful suggestions. We will incorporate the new analysis into Supplementary Section §C, which focuses on the theoretical study of the adapted feature space. We believe this addition offers a valuable complementary perspective and further deepens the conceptual understanding of our approach.

---

> ### Comment · Reviewer_6sqR · 2025-08-05
>
> Thanks authors for the detailed and comprehensive rebuttal. The additional experiments and analysis addressed my concerns. I would like to increase my score to 5.

---

> > ### Author Response · Authors · 2025-08-05
> >
> > We sincerely thank the Reviewer for the kind and positive feedback. We’re happy that our additional analyses helped clarify the key points, and we deeply appreciate the time and care dedicated to reviewing our work. We will incorporate all related discussions and analyses into the final version of the paper.

---

### Official Review · Reviewer_vipR · 2025-07-06

**Clarity:** 3
**Significance:** 3
**Originality:** 3
**Rating:** 5
**Confidence:** 4

**Summary:**

This work aims to solve Few-shot semantic segmentation by leveraging SAM-2 built in feature matching matching process. It argues that SAM2 encodes rich semantic structure, however representations are entangled with object tracking cues and cannot be naively extended for few-shot semantic segmentation. It proposed SANSA framework that learns a simple adaptor on top of SAM features that repurposes SAM-2 for few-shot semantic segmentation with minimal samples. SANSA achieves state-of-the-art results on standard benchmarks and remains faster and more compact than existing approaches.

**Questions:**

I have two major questions -
1. Do we really need masks? With SAM2 strong segmentation ability one doesnt need fully fine-grained mask for the few shot samples.
2. Is the performance gain due to stronger SAM2? Prior works like VRP-SAM are based on older SAM and hence maybe limited in terms of performance.

**Ethical Concerns:**

["NO or VERY MINOR ethics concerns only"]

**Final Justification:**

Thank you so much for the rebuttal! It answers all my concerns. And I propose a score of 5. I really like the analysis of how SAM2's inherent feature matching can be extended to few shot segmentation. The work is well motivated and a nice read with sufficient experiments and ablations.

**Limitations:**

One limitation can be -

The work only segments single object classes and maynot be straightforward to extend to multi object segmentation?

**Quality:**

3

**Strengths And Weaknesses:**

Strengths -

1. The idea of extending SAM-2 which is based on video object segmentation for few shot segmentation is interesting and novel. While it is not straightforward to extend SAM-2 for few shot segmentation, it does have inbuilt feature matching property which is exploited by this work.

2. The feature visualisation in Fig 4 showing better and separated semantic class features further improves the claim of better features with proposed approach. It even shows better part segmentation when the work performs k-means clustering the object features.

3. The work is well written and easy to follow. It achieves state of the art performance on standard benchmarks (tab 1) with complete ablations and also compares performance with in-context learning methods (tab 2).


Weaknesses -

1. Performance with just prompts/ Do we really need masks? - The proposed work uses segmentation masks for training in few shot manner. However SAM-2 can accept just prompts such as points and generate nice segmentation masks. How does the performance change when just prompts are provided in few shot samples (lets say just points instead of masks). Having full binary masks doesnt seem to be necessary.

2. Integration of negative prompt - Did the work also trying integration of negative prompts? The negative prompt might further improve representation quality for desired object and overall improve performance.

3. Performance gain due to stronger SAM2? Prior works like VRP-SAM, PerSAM are based on SAM (previous version of SAM-2) which supposedly is inferior. Its unclear if the performance gain is based on stronger SAM-2 or by the proposed method.

4. Single class segmentation? Is the work limited to segmenting a single class? Can it segment multiple class objects in a single image? This should be discussed in the work.

---

> ### Author Rebuttal · Authors · 2025-07-29
>
> We thank the Reviewer for the thoughtful and positive feedback. We are glad that the proposed solution was found to be both **interesting** and **novel**, and that the **feature analysis**, **comprehensive ablations**, and **state-of-the-art results**, across standard benchmarks and in-context settings, were recognized as strengths of the work. We also sincerely value the constructive questions raised. In the following, we did our best to provide clarifications and additional analyses to address these points as clearly and thoroughly as possible.
>
> --------------
>
> 1.
> > **W1,Q1:** *"Do we really need masks? With SAM2 strong segmentation ability one doesnt need fully fine-grained mask for the few shot samples.  How does the performance change when just prompts are provided in few shot samples (lets say just points instead of masks)?*
>
> We thank the reviewer for this valuable question. We fully agree that reducing annotation effort is desirable in practical applications. Motivated by this, our main paper includes an evaluation of SANSA under different prompt types (*points*, *boxes*, and even *scribbles*), reported in **Figure 6 (top table), reproduced below**. Implementation details are provided in **§F.3 of the Supplementary**, along with extensive qualitative results in **Figure 5 of the Supplementary Material**.
>
> As shown below, SANSA achieves **state-of-the-art performance across all prompt types**, significantly outperforming VRP-SAM, also targeting *promptable* few-shot segmentation, by **+15.0%** with points, **+5.8%** with scribbles, and **+4.6%** with boxes. Furthermore, SANSA exhibits strong robustness, with a performance drop of –6.7% from full-mask to point supervision, compared to –15.5% for VRP-SAM.
>
> We also highlight that several **recent works** [35, 67, 71] built on SAM **cannot operate with point- or box-level prompts**, as they **rely on external encoders** (*e.g.*, DINOv2) for feature matching, and use SAM to produce the segmentation mask given the matched coordinate locations. VRP-SAM purposefully trains an additional decoder to support such prompts. In contrast, SANSA is the first to directly adapt the SAM2 feature space so that *the same features* can drive both feature matching and segmentation, enabling to **fully repurpose** SAM2 promptability for few-shot segmentation with low-effort prompts (L255-L264).
>
>
> **Table from Figure 6 (top table) of the Main paper**
> | Method        | Params | Point | Scribble | Box  | Mask |
> |---------------|--------|:-----:|:--------:|:----:|:----:|
> | VRP-SAM       | 670M   | 38.4  | 47.3     | 49.7 | 53.9 |
> | **SANSA**  | **234M**   | **53.4 (+15.0)** | **53.1 (+5.8)** | **54.3 (+4.6)** | **60.1 (+6.2)** |
>
> -----------------
>
> 2.
> > **W3,Q2:** *Is the performance gain due to stronger SAM2? Prior works like VRP-SAM are based on older SAM and hence maybe limited in terms of performance.*
>
> We agree with the Reviewer on the importance of proper baselines. We address this point in **Table 4 of Section §E of the Supplementary Material, reproduced below**, where we isolate the effect of upgrading from SAM to SAM2 in recent Segment Anything-based methods. Recent state-of-the-art [21,20,42] models adopt a modular design: they use **external encoders for feature matching** (e.g. DINOv2), and treat **SAM as a promptable segmentation decoder** operating at the **image level**.
> As shown in Table 1, replacing SAM with SAM2 results in minimal performance gains (< 2% mIoU). This is consistent with results reported in the original SAM2 paper [27], which was designed for *video* segmentation and reports similarly small improvements in promptable *image-level* segmentation.
>
> These findings demonstrate that our performance improvements do not arise from simply using SAM2. Instead, the key novelty lies in **how SAM2 is used**: our method adapts the feature space, revealing a previously **unexplored semantic structure** that supports **both matching and segmentation** within a unified framework.
>
>
> **Table 4 from Section §E of the Supplementary Material**, † Newly added for rebuttal
> | Method          | Feature Matching Backbone | Segmentation Model | COCO-20i mIoU(%) | LVIS-92i  mIoU(%)  |
> |-----------------|----------------------------|--------------------|-------------------|-------------------|
> | PerSAM    | –                          | SAM / SAM2         | 23.0 / 23.6 †       | 11.5 / 13.2 † |
> | Matcher    | DINOv2                     | SAM / SAM2         | 52.7 / 52.4       |  33.0 / 33.6 † |
> | VRP-SAM    | ResNet-50                  | SAM / SAM2         | 53.9 / 54.3       |  28.3 / 28.9 † |
> | GF-SAM    | DINOv2                     | SAM / SAM2         | 58.7 / 58.9       |  35.2 / 37.0 † |
> | **SANSA (ours)**| –                          | **SAM2**            | **60.1**          |  **49.8** |
>
> ------------------------------
>
> 3.
> > **W3, L1:** *Is the work limited to segmenting a single class?*
>
> We thank the reviewer for raising this point which gives us the opportunity to further clarify the discussion provided in Section §A.1 of the Supplementary. Our model **can perform multi-class segmentation**. Indeed, SANSA builds upon SAM2, which already supports tracking and segmentation of multiple objects by storing multiple object embeddings in the Memory Bank. In the same way, our framework naturally handles multiple categories by storing separate reference representations, predicting each class mask independently, and aggregating them into a final segmentation map [27].
>
> That said, the current setup is **shared by all existing methods** [53, 35, 67], where categories are **segmented independently**, without leveraging potential relationships between them. In the Section §A.1 of the Supplementary, we highlight this as a **broader limitation of the current few-shot segmentation paradigm**. This paradigm is **effective for evaluating per-class generalization**; however, exploring how to incorporate cross-category reasoning could be a promising direction for future research.
>
> --------
>
> 4.
> > **W2**: *Did the work also trying integration of negative prompts?*
>
> We thank the reviewer for this insightful suggestion. While our main experiments do **not** include negative prompts, so **as to ensure a fair comparison** with prior works, we find this direction **highly compelling** and took the **opportunity to explore it further**. The key challenge lies in adapting the notion of *negative prompting*, originally designed for interactive segmentation, to the setting of few-shot segmentation.
>
> Delving into this analysis, a key distinction is that negative prompts in the SAM2 paper are geometric cues (*e.g.*, clicks) provided at test time to correct segmentation mistakes. Inspired by the reviewer suggestion, we propose to broaden the concept of negative prompting by introducing **semantic negative prompts**, namely, annotated examples of visually similar but semantically different categories, provided alongside the reference as contrastive context  (*e.g.*, including a photo of a *horse* to help disambiguate the target class *zebra*).
>
> We thus explored two main strategies: geometric negative prompts (for test-time correction), and semantic negative prompts (for improved reference conditioning). Our experiments include both training-free and lightweight training-based variants, and are summarized below:
>
> - (Training Free) **Geometric Negative Prompts**: In this setting, negative clicks are provided at test time on the target image, mimicking interactive correction. Since SANSA reuses SAM2 prompt encoder, we can directly leverage SAM2 built-in negative point mechanism.
> - **Semantic Negative Prompts**: We then explore a novel interpretation of negative prompting in FSS, providing an annotated reference image of a semantically different (but potentially confusing) category. We implement it in two variants:
>     - *Training Free Negative Prompts*: We encode the negative reference into SAM2 memory bank, assigning it an empty mask. This signals to the model that this region contains no object matching the current task.
>     - *Training-based Negative Prompts*: We extend SAM2 prompt encoder with a small MLP that encodes the average feature of a negative object into a learnable negative prompt token, similar to SAM2 handling of negative clicks.
>
>
> **Table R1: Geometric vs. Semantic Negative Prompts**
> | Method                                 | mIoU |
> |----------------------------------------|:----:|
> | **SANSA (baseline)**                   | 60.1 |
> | **Geometric Negative Prompt**          |      |
> | &nbsp;&nbsp;&nbsp;&nbsp;• +1 click     | 60.8 |
> | &nbsp;&nbsp;&nbsp;&nbsp;• +3 clicks    | 62.9 |
> | &nbsp;&nbsp;&nbsp;&nbsp;• +5 clicks    | 64.5 |
> | **Semantic Negative Prompt**           |      |
> | &nbsp;&nbsp;&nbsp;&nbsp;• Training-Free Negative Prompts      | 61.2 |
> | &nbsp;&nbsp;&nbsp;&nbsp;• Training-Based Negative Prompts | 63.0 |
>
>
>
> We believe this analysis to be highly interesting and potentially impactful. However, we emphasize that it opens a **broad set of questions**, particularly for semantic negative prompting, for which **standardized baselines are currently lacking**. We plan to explore these directions more rigorously in **future works**.
>
> ---
>
> Once again, we sincerely thank the Reviewer for their time and careful consideration. We hope our responses have provided the necessary clarifications and addressed all points raised.

---

> > ### Comment · Reviewer_vipR · 2025-08-04
> >
> > Thank you so much for the nice rebuttal. All my concerns are addressed in the rebuttal. It would be really if the supplement includes the discussion and results presented in this rebuttal, particularly W1, W2 and W3. I have increased my score 5.

---

> > > ### Author Response · Authors · 2025-08-05
> > >
> > > We sincerely thank the Reviewer for the kind and positive comment! We're very glad to hear that the rebuttal addressed all the concerns raised. We would also like to take this opportunity to express our gratitude once again for the time and care dedicated to reviewing the work. We will make sure to integrate the suggestions from the rebuttal into the final version.

---

### Comment · Area_Chair_t3Le · 2025-08-04
**Please Read the Rebuttal and Discuss**

Dear Reviewers 6sqR, Cxfs. vipR

The authors have submitted their rebuttal.

Please carefully review all other reviews and the authors’ responses, and engage in an open exchange with the authors.

Kindly post your initial response as early as possible within the discussion window to allow sufficient time for interaction.

Your AC

---

### Note · Authors · 2025-08-13

We sincerely thank the AC and Reviewers for their engagement and constructive feedback throughout the review process. We are grateful that the additional analyses and clarifications in our rebuttal fully addressed all concerns.


In the final version, if the paper were to be accepted, we will integrate all suggestions from the discussion phase, including:

- **Improved Clarity** :
   - Improve caption description and comparisons in Figure 1 and explanation of PCA color encodings in Figure 2.
   - Clearer reference to analysis in the Supplementary (*e.g.* baselines isolating the role of SAM2 vs. SAM) and main paper (*e.g.* low-effort point/box prompts).

- **Extended Analysis**:
    - Adapter placement ablation (**new Tab. R4**), negative prompt experiments (**new Tab. R1**) and spatial continuity analysis (**new Tab. R2**).
    - Expanding the discussion on *fine-tuning vs. adaptation* comparison (Supp. §C.3) with two additional configurations: full fine-tuning from SAM2 weights and training from scratch (**new Tab. R3**).
    - Expanding ablation on hyperparameter *J* (**new Tab. R5**) to validate mask propagation stability.
- **Reproducibility**:
    - Expanding Supp. §G.1 to include the training hyperparameter *k=1* and to clarify that training episodes construction follows the *standard* few-shot segmentation protocol.

We thank once again the Reviewers and AC for their time and consideration, and look forward to refining the final version to reflect all feedback from the review process.

---

### Decision · Program_Chairs · 2025-09-17

**Decision:**

Accept (spotlight)

**Comment:**

This paper exploits the latent semantic structure within SAM2 for few-shot segmentation.

While the reviewers like the idea and results in general, reviewers raised concerns about novelty, theoretical depth, and fairness of comparisons. In particular
- Reviewer eWJd felt the contribution was incremental (applying AdaptFormer to SAM2), and requested more systematic ablations (adapter placement, hyperparameters)
- Reviewer 6sqR commented that there lack of a theoretical analysis about the adaptation of SAM2 to spatially independent inputs
- Reviewer vipR questioned whether performance gains were mainly due to stronger SAM2 compared to prior SAM-based methods

The rebuttal convincingly addressed these through additional experiments and clarifications. Why not higher scores:
- Even after the rebuttal clarified the novelty (semantic structure hidden in SAM2), the perception that this was more an adaptation than a fundamentally new algorithm limited enthusiasm.

Reviewers converged on 5 as the appropriate score. The consensus is that SANSA is a valuable and timely contribution that will be of interest to both the segmentation and foundation model communities.